# Circum-Arctic release of terrestrial carbon varies between regions and sources

Jannik Martens [1,2,3], Birgit Wild [1,2], Igor Semiletov[4,5,6], Oleg V. Dudarev[4,6] & Örjan Gustafsson [1,2] ✉

Arctic change is expected to destabilize terrestrial carbon (terrOC) in soils and permafrost, leading to fluvial release, greenhouse gas emission and climate feedback. However, landscape heterogeneity and location-specific observations complicate large-scale assessments of terrOC mobilization. Here we reveal differences in terrOC release, deduced from the Circum-Arctic Sediment Carbon Database (CASCADE) using source-diagnostic ($\delta^{13}$C-$\Delta^{14}$C) and carbon accumulation data. The results show five-times larger terrOC release from the Eurasian than from the American Arctic. Most of the circum-Arctic terrOC originates from near-surface soils (61%); 30% stems from Pleistocene-age permafrost. TerrOC translocation, relative to land-based terrOC stocks, varies by a factor of five between circum-Arctic regions. Shelf seas with higher relative terrOC translocation follow the spatial pattern of recent Arctic warming, while such with lower translocation reflect long-distance lateral transport with efficient remineralization of terrOC. This study provides a receptor-based perspective for how terrOC release varies across the circum-Arctic.

The Arctic is warming twice as fast as the global average[1], which causes destabilization of high-latitude soils and other permafrost deposits. These systems contain terrestrial organic carbon (terrOC) stocks equivalent to twice the carbon pool in the atmosphere. Arctic permafrost soils hold about 1000 Pg OC in the top 3 m[2]. Another 200 Pg OC are stored in old Pleistocene-age permafrost such as coastal Ice Complex Deposits (ICD)[3] that occur primarily in northeast Siberia and Alaska, and are particularly prone to collapse[4,5]. Furthermore, peatlands in boreal soils outside the permafrost zone are estimated to contain an additional 230 Pg OC[6]. Climate warming may cause release of terrOC from these pools by, e.g., deepening of the seasonally thawed active layer of permafrost soils, thermokarst, thermal collapse and erosion of ice-rich ICD, or degradation of peatlands. This may expose large amounts of terrOC to fluvial transport and to microbial degradation, with subsequent emissions of greenhouse gases (foremost $CO_2$ and $CH_4$), climate-carbon feedback[6–8] and Arctic ocean acidification[9]. However, large uncertainties exist regarding the dynamics and processes of large-scale terrOC release in the Arctic.

Robust projections of the future Arctic carbon-climate feedback require a quantitative understanding of large-scale terrOC release. Yet, the overarching picture of terrOC release across the circum-Arctic rests on a scattered collection of location-specific observations. State-of-the-art permafrost models retain large uncertainties of (i) permafrost area, (ii) depth of the active layer, (iii) frequency and spatial extent of thermokarst; (iv) net carbon exchange between terrOC systems and the atmosphere, as well as (v) lateral leakage of carbon to aquatic conduits. Overall, non-linear terrOC release pathways are particularly challenging to predict, such as the collapse of ICD, thermokarst formation, tundra and forest fires, as well as fluvial release of OC; processes that may all affect the net ecosystem carbon balance[6,7,10,11].

Lateral leakage of terrOC through aquatic conduits is a key component of the carbon cycle in the often water-logged circum-Arctic.

[1]Department of Environmental Science (ACES), Stockholm University, Stockholm, Sweden. [2]Bolin Centre for Climate Research, Stockholm University, Stockholm, Sweden. [3]Lamont-Doherty Earth Observatory of Columbia University, New York, USA. [4]Il'ichov Pacific Oceanological Institute FEB RAS, Vladivostok, Russia. [5]Higher School of Economics (HSE), Moscow, Russia. [6]Tomsk State University (TSU), Tomsk, Russia. ✉ e-mail: orjan.gustafsson@aces.su.se

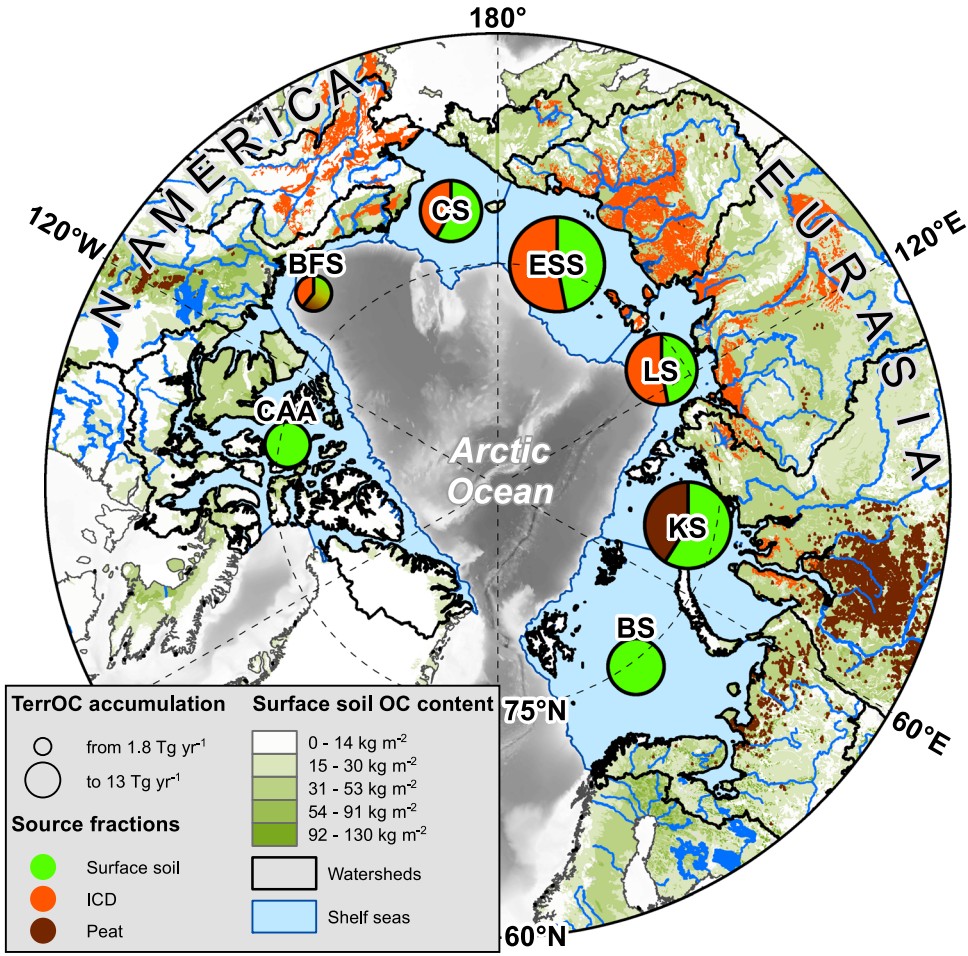

**Fig. 1 | Overview of the different terrestrial organic carbon (terrOC) sources to circum-Arctic shelf sediments.** The terrOC sources include surface soil (SurfSoil-OC; green colors), Ice Complex Deposits (ICD-OC; orange colors) and peat (brown colors), while petrogenic OC is excluded from terrOC. The size of the pie charts is proportional to the relative contributions of the different sources to the terrOC accumulating in the recipient shelf sediments, (ranging from 1.8 Tg yr⁻¹ in the BFS to 13 Tg yr⁻¹ in the ESS). Green shades on land indicate surface soil OC concentrations[22,23], orange shades show the distribution of ICD[3] and brown areas indicate peatlands[6]. Blue-outlined shapes show the seven circum-Arctic shelf seas while black delineations indicate the corresponding drainage basins. CAA Canadian Arctic Archipelago, BFS Beaufort Sea, CS Chukchi Sea, ESS East Siberian Sea, LS Laptev Sea, KS Kara Sea, and BS Barents Sea. The Arctic Ocean base map is based on IBCAOv4[58,59].

The large rivers and coastal shelf seas act as nature's own integrator and permit studies of terrOC release from different source compartments as well as geographically distinct drainage basins and coastal regions[12–15]. Arctic rivers may shed light on terrOC release on a drainage basin scale and resolve the seasonal variation of different source compartments[16,17]. By contrast, the seven Arctic shelf seas have the advantage to serve as natural integrators of terrOC release from the river drainage basins through the sequestration of riverine OC from dissolved and particular forms in their sediments upon aggregation and settling with increasing salinity[18]. Moreover, shelf seas are recipient of OC from erosion of coastal permafrost deposits (ICD), which is suggested to be the dominating vector of terrOC release to the extensive Laptev and East Siberian Seas[4,5,19].

The present study leverages off the recently established Circum-Arctic Sediment Carbon DatabasE (CASCADE)[20,21] and uses an inverse receptor-based approach to deduce terrOC release from different source compartments in the circum-Arctic regions (Fig. 1). The CAS-CADE provides OC concentrations for over 4000 locations across the Arctic Ocean shelf seas. Dual-isotope data (δ¹³C; Δ¹⁴C; $n = 260$ locations) allows to calculate the relative contribution of different sources to the terrOC sequestered in the sediments. Further, fluxes of terrOC release from different terrestrial compartments are quantified using ²¹⁰Pb-based mass accumulation rates ($n = 164$) in the circum-Arctic

shelf seas. Finally, the relative vulnerability of the different terrOC compartments to be released was investigated by comparing shelf terrOC accumulation fluxes with the corresponding catchment-specific terrOC stocks; i.e., OC in permafrost[22] and non-permafrost surface soils (SurfSoil)[23], ICD[3], and peat[6]. To this end, an Integrated Carbon Release Index is introduced (I-CRI) to contrast geographical and source-specific differences of terrOC release across the circum-Arctic.

## Results and discussion
### Dual-isotope source apportionment in the circum-Arctic
The dual-isotope information (δ¹³C/Δ¹⁴C) provided by CASCADE allows us to distinguish between different terrOC sources and OC produced by marine phytoplankton (Fig. 2). Given the heterogeneity of terrOC compartments around the circum-Arctic, this study employs different terrOC end members for each shelf sea (Supplementary Table 4), in each case based on large underlying end member collections of data on δ¹³C and Δ¹⁴C (Supplementary Text 2). Accordingly, OC from surface soils to a maximum depth of 100 cm was applied as end member in the entire circum-Arctic, with the exception of the Beaufort Sea where also deeper layers and peatlands were included due to overlapping isotopic compositions (δ¹³C/Δ¹⁴C) and limitation to three end members in the mixing model. Pleistocene age OC in ICD occurs in

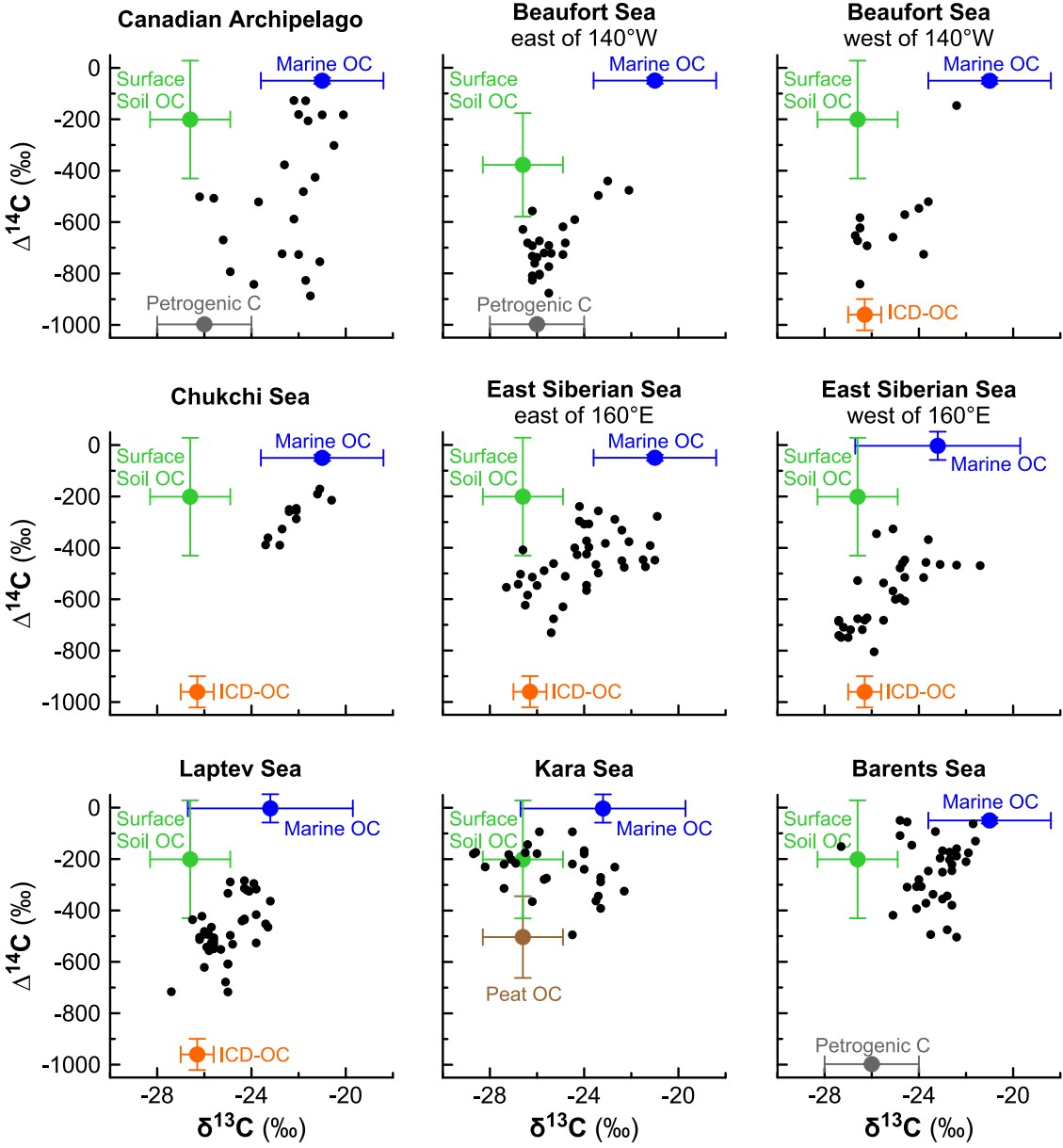

**Fig. 2 | Scatter plots of δ¹³C-Δ¹⁴C patterns of OC in circum-Arctic shelf sediments based on the Circum-Arctic Sediment Carbon Database - CASCADE[20,21].** Shown are data for surface sediment samples, as well as the mean and standard deviations of the different end members shown as colored points and error bars (in green Surface Soil OC, in orange Ice Complex Deposit – ICD-OC, in brown Peat OC, in blue Marine OC and in gray Petrogenic C) for the circum-Arctic shelf seas. Note that the Beaufort and East Siberian Seas are each divided into two end member systems due to the absence of ICD-OC east of 140°W (Beaufort Sea) and the use of a wider Marine OC end member west of 160°E (East Siberian Sea). The selection of the end members is further detailed in the Methods and Supplementary Methods 1.

northeastern Siberia, Alaska and western Canada, and was hence considered as second terrOC end member in the Laptev, East Siberian, Chukchi and western Beaufort Sea shelves. Further, deep peat below 100 cm depth was considered as pre-aged terrOC end member for the Kara Sea, which has the world's largest peatland in its catchment. Petrogenic OC released from rock weathering was used as ¹⁴C-depleted terrOC end member in the Canadian Arctic and the Barents Sea, to account for the significant release of petrogenic OC in these regions[24,25] (Fig. 2). Each end member is described by the mean ± standard deviation (s.d.) of the underlying data collection, which represents the natural variability of the different source compartments and is mirrored in the uncertainty of the resulting source fractions.

TerrOC end members were also corrected for aging during cross-shelf transport. Cross-shelf aging of terrOC was previously quantified by ¹⁴C dating of terrestrial organic compounds[26], which is here applied

to correct for the transport distance of terrOC at each sampling location of the dataset (Methods, Cross-shelf transport correction). By contrast, the ¹³C end member was assumed to stay constant as previous studies indicated continued terrestrial ¹³C signatures despite major cross-shelf aging of terrOC[26]. Previous research also suggested aging of marine OC during cross-shelf transport[27–29]. However, the source location and the transport dynamics of marine OC are uncertain and the scale for marine OC aging in continental shelf seas is unknown, albeit likely much less than for terrOC due to both longer transports distances and lower recalcitrance.

## Circum-Arctic features of terrestrial carbon release

The large underlying CASCADE dataset offers a circum-Arctic perspective and reveals that terrOC release in the Eurasian Arctic sector is much larger than in the North American sector. The CASCADE-based

terrOC sequestration in the respective shelf sea receptor sediments reveals that the East Siberian Sea drainage basin accounts for 28% of the total circum-Arctic terrOC flux, while the Kara Sea accounts for 24% and the Laptev Sea for 16% (Fig. 1; Supplementary Table 1). Including also the Barents (10%) and Russian Chukchi seas (6%), five times more terrOC is received in the Eurasian-Arctic sector than in the American-Arctic sector. This partitioning is broadly consistent with the hemispheric proportions of fluvial OC discharge and land-based terrOC stocks, which both are much larger for the Eurasian than the American watersheds of the Arctic Ocean. Consistent with this large-scale assessment, Eurasian rivers account for about 83% of the total fluvial OC input to the Arctic Ocean[30,31]. Moreover, the partitioning of terrOC release correlates with the relative distribution of the land-based terrOC stock in the circum-Arctic ($R^2 = 0.56$; $p = 0.05$; Supplementary Fig. 1i)[2,3,6], with 78% of the SurfSoil-OC and 83% of the total peat OC being located within the Eurasian drainage basin[6,22,23]. Although the dual-isotope receptor approach holds uncertainty, these results support the use of shelf sediments to deduce large-scale features of terrOC release and underscore the importance of the Eurasian-Arctic system for terrOC vulnerability in the Arctic.

The CASCADE facilitates also comparison of the propensity of different terrOC sources to be released and to investigate differences in this across the circum-Arctic. Dual-isotope ($\delta^{13}C/\Delta^{14}C$) source apportionment of OC reveals that most of the terrOC was released from surface soils (SurfSoil $61 \pm 39\%$; Fig. 1). The relative contribution of SurfSoil-OC varies regionally, with about half of the terrOC in the East Siberian ($47 \pm 29\%$) and Laptev seas ($47 \pm 30\%$), and $59 \pm 39\%$ in the Kara Sea. The remaining terrOC originates from ICD (East Siberian and Laptev Seas) and deep peat deposits (Kara Sea; absolute fluxes are further discussed in Supplementary Discussion 1). About a third of the terrOC was released from ICD ($30 \pm 10\%$), which is characteristic for the coastline in Northeast Siberia, Alaska, and Western Canada. Peat deposits contributed a total of $10 \pm 6\%$ to the circum-Arctic terrOC accumulation in sediments, with a regional maximum of $41 \pm 24\%$ of the total terrOC in the Kara Sea catchment. In the Beaufort Sea, the combined release of SurfSoil and Peat-OC accounts for $61 \pm 40\%$ of the total terrOC release (Fig. 1). The circum-Arctic release pattern of the different terrOC sources broadly resembles its distribution pattern on land (Fig. 1)[2,3,6], which supports the suitability of the receptor-based approach[4,32,33].

The large portion of terrOC released from ICD emphasizes the particular importance of collapsing Arctic coastlines in northeastern Siberia, Alaska and the US Beaufort Sea (Fig. 1). While ICD is present in the catchments of only four out of the seven shelf seas (Supplementary Text 2)[3], it contributes a third of the total terrOC release for the whole circum-Arctic (Supplementary Table 1), and accounts for $53 \pm 17\%$ of the total terrOC flux in the Laptev and East Siberian Sea systems. The highest input of ICD-OC to the northeastern Siberian seas likely relates to severe coastal erosion of up to 5 m yr$^{-1}$, high coastal cliffs, and large volumes of ground ice in this area (Fig. 3)[34]. Release of ICD-OC also occurs inland (e.g. by riverbank erosion) and routes via rivers[10,14,17]. Yet, Pleistocene-aged permafrost OC accounts for less than a fifth of the dissolved and particulate OC export from, e.g., the Lena and Kolyma rivers that drain extensive ICD landscapes[17], which points at a stronger coupling of ICD-OC release with erosion of the vast and exposed coastline. Based on a large-footprint dataset, the present study suggests that coastal erosion drives the large fluxes of ICD-OC release in the circum-Arctic.

In addition to terrOC from soils, permafrost, and peat, petrogenic rocks release considerable amounts of OC in the catchments of the Canadian Arctic and off Svalbard. For the Canadian Arctic Archipelago and the Barents Sea, petrogenic OC accounted for $37 \pm 26\%$ and $16 \pm 11\%$ (mean $\pm$ s.d.) of their total OC releases, respectively. For the Beaufort shelf, we find $51 \pm 13\%$ of the OC in sediments to originate from petrogenic sources, which agrees with the large range (19–88%)

suggested by previous studies specifically addressing this region[24,25,35,36]. Further, the concentrations of petrogenic OC in Beaufort Sea sediments ($0.66 \pm 0.18\%$; mean $\pm$ s.d.) are also similar to the range of petrogenic OC concentrations in fluvial suspended material from within the Mackenzie basin (0.12–0.63%)[35,36], which thus is broadly consistent with the results of the dual-isotope source apportionment and indicates that petrogenic OC in Beaufort Sea sediments is largely attributed to rock weathering and river export from the Mackenzie river and its tributaries[36]. Less certain, however, is whether petrogenic OC−likely to be highly recalcitrant−contributes to any notable emissions of $CO_2$ and thus affects the active carbon cycle, which is why petrogenic OC contributions were not considered terrOC release in the forthcoming part of the present study.

## Comparison of terrestrial carbon release with circum-Arctic carbon stocks

Terrestrial OC accumulation in the recipient shelf seas can be compared with the vast terrOC stocks of the circum-Arctic source compartments to provide a metric and perspective on regional differences in large-scale terrOC releases. To this end, the Integrated Carbon Release Index (I-CRI) represents the percentage of the total terrOC, SurfSoil-OC, ICD-OC, and Peat-OC stock in each drainage basin (Supplementary Table 2)[3,6,22,23] that accumulates in sediments over the course of 100 years (Supplementary Table 3), where the estimate of the total uncertainty (s.d.) includes the spatial variability of OC accumulation fluxes, as well as the uncertainties of the source apportionment and that of the land-based terrOC stock estimates. For total terrOC, the results exhibit large differences between the geographical regions, from the lowest relative release in the Beaufort Sea basin (I-CRI$_{terr}$ $0.2 \pm 0.1$, % terrOC translocated 100 yr$^{-1}$) to five times higher relative release in the basins of the Canadian Arctic Archipelago ($1.0 \pm 0.7$) and the East Siberian Sea ($1.0 \pm 0.4$; Fig. 3; Supplementary Table 3). The dynamic range is yet larger for the source-compartment-specific I-CRI$_{SurfSoil}$, ranging from $0.2 \pm 0.1$ in the Beaufort Sea to $1.4 \pm 0.9$ in the East Siberian Sea, suggesting that relative OC release from surface soils varies by a factor of seven in the circum-Arctic. By contrast, the I-CRI$_{ICD}$ reveals a smaller range from $0.4 \pm 0.2$ in the Chukchi Sea to about two times higher values in the US Beaufort Sea ($1.1 \pm 0.3$), and in the East Siberian Sea ($0.8 \pm 0.3$). The I-CRI clearly shows differences in terrOC release propensities across the seven Arctic shelf sea catchments and between the terrOC source compartments.

Comparing the I-CRI with environmental properties of the circum-Arctic drainage basins offers an approach to explore possible drivers behind regional differences in terrOC release. First, the variability in I-CRI$_{terr}$ and I-CRI$_{SurfSoil}$ resembles the spatial pattern of catchment-specific summer warming trends for the period 1960-2015 (Fig. 4a; Supplementary Fig. 1c, d)[37,38]. At the same time, terrOC release from surface soils is consistently higher in small shelf catchments with coverage of mostly high-latitude areas (I-CRI$_{SurfSoil}$ > 1; Chukchi, East Siberian, Barents Seas and the Canadian Arctic Archipelago) than in larger drainage basins that extend further to the south (Beaufort, Laptev and Kara Seas; I-CRI$_{SurfSoil}$ < 0.4; Fig. 4b; Fig. Supplementary Fig. 1a). In addition, the I-CRI$_{ICD}$ correlates with the regional pattern of average coastal erosion in the ICD-hosting shelf sea catchments (Supplementary Fig. 1b), with average erosion rates of 1.1 and 0.9 m yr$^{-1}$ in the East Siberian and Beaufort Seas when compared with 0.7 and 0.4 m yr$^{-1}$ for the Laptev and Chukchi Seas[34] ($R^2 = 0.92$; $p = 0.04$). We hypothesize that the large differences in relative terrOC release to the Arctic Ocean receptors reflect a combination of processes, including regional differences of Arctic climate change, degradation during transport via hydrological conduits, and the thermal state of the ground. Key hypothesized mechanisms are further addressed below.

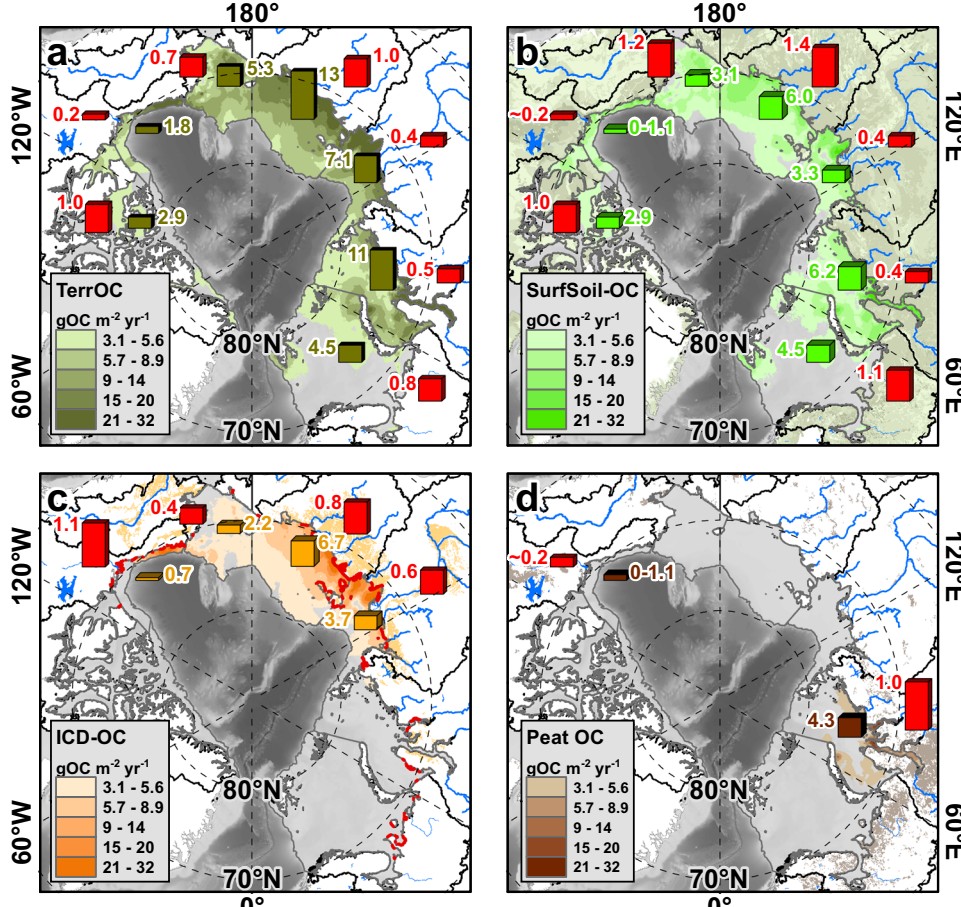

**Fig. 3 | Circum-Arctic patterns of terrestrial organic carbon (terrOC) release.** The four maps show accumulation rates of released terrOC apportioned between **a** total terrOC in dark green shades; **b** specifically surface soil (SurfSoil) incl. permafrost active layer in light green shades; **c** Ice Complex Deposits (ICD) in orange shades[3] incl. costal erosion with >1 m yr[−1] outlined in red[34]; and **d** Peat OC in brown colors. Furthermore, the total flux (in Tg yr[−1]) is shown for each shelf sea as colored bar charts, together with red bars indicating the Integrated Carbon Release Index (I-CRI) for each source compartment. The I-CRI is a relative measure of terrOC recipient flux, relative to source region stock, from different compartments (see main text). Arctic Ocean base maps are based on IBCAOv4[58,59].

## Potential effects of recent climate warming on carbon release

The amplified rise in Arctic temperatures may be an important driver of terrOC release from circum-Arctic permafrost soils. For the period 1960–2015, circum-Arctic average summer temperatures (May-October) increased by 1.4 °C in the Beaufort Sea catchment and up to 2.2 °C in the Canadian Arctic Archipelago[37,38]. This caused notable warming of permafrost soils, particularly those situated in the continuous permafrost zone[39]. One of the effects of Arctic change is the deepening of the seasonal permafrost active layer, which is suggested by an increasing number of yet-scattered records[8,40,41]. Furthermore, first signs of accelerated coastal erosion in the circum-Arctic have been reported[42,43].

Geographical variability in the temperature increase across the circum-Arctic might contribute to the regional differences in terrOC release. This is supported by the spatial correlation of the I-CRI$_{terr}$ ($R^2 = 0.86$; $p = 0.003$) and I-CRI$_{SurfSoil}$ ($R^2 = 0.54$; $p = 0.059$) with the average summer warming trends 1960–2015 in the circum-Arctic (Fig. 4c; Supplementary Fig. 1c, d). These observations are also broadly in line with the scattered observational dataset of active layer development, with significant deepening in, e.g., northeastern Siberia over the past decades[8]. Further, a biogeochemical model suggests that the largest increase of OC release during the past century was in the high-latitude Canadian Arctic Archipelago (22%), and in the drainage basins of the East Siberian (19%) and Chukchi seas (incl. Bering Strait; 42%)[44], which is also supported by the

increased discharges of rivers in northeastern Siberia[45,46]. While this may suggest a connection between terrOC release and recent warming trends across the circum-Arctic, the range in warming is only 50% while the range in I-CRI is a factor of 5–7. In addition, recent Arctic warming, terrOC release and translocation may operate over different time scales. It therefore seems unlikely that temperature increase alone can explain the large differences in terrOC release across the circum-Arctic pinpointed in this study.

## Potential effects of degradation during fluvial transport on carbon release

A portion of terrOC released from surface soils, peat or other deposits is generally remineralized during transport within the catchment area[47]. This may be particularly anticipated for the circum-Arctic, where large parts of the catchment area are water-logged, and freshwater runoff governs terrOC release and degradation[10]. The I-CRI of the different shelf seas varies by a factor of five for total terrOC (I-CRI$_{terr}$), and by a factor of seven for surface soils (I-CRI$_{surfsoil}$), showing generally higher source-receptor fluxes for smaller high-latitude basins, and lower fluxes in larger, southward-extending, catchments. Hence, the I-CRI is spatially correlated with the southward extent ($R^2 = 0.68$ and $p = 0.022$ for I-CRI$_{terr}$; $R^2 = 0.67$ and $p = 0.025$ for I-CRI$_{surfsoil}$; Supplementary Fig. 1g, h), which may reflect efficient underway degradation and significant loss of organic material of distantly sourced organic matter.

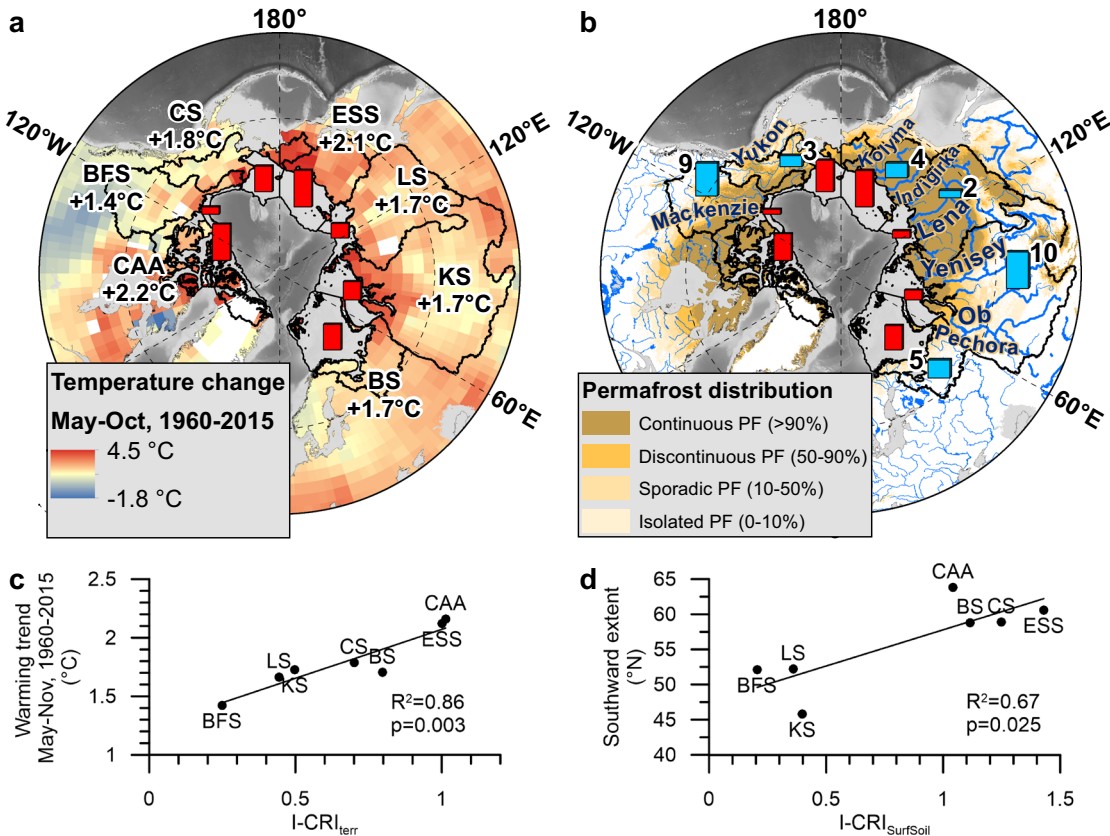

**Fig. 4 | Possible drivers of terrestrial organic carbon (terrOC) release in the circum-Arctic.** The map on the left (**a**) shows the change of the annual summer temperatures (May-Oct) for the period 1960-2015[37,38], together with the Integrated Carbon Release Index (I-CRI) for total terrOC as red bars. On the bottom left (**c**), the correlation between the warming trend and the I-CRI$_{terr}$ is shown. The map on the right (**b**) shows the circum-Arctic drainage basins incl. its major river systems and the Arctic permafrost (PF) zones[49], next to the I-CRI for surface soil OC as red bars

and the ratio of watershed-internal $CO_2$ evasion to the lateral export of terrOC via the major Arctic rivers as blue bars[31,50–52]. On the bottom right, **d** the correlation between the southward extent of the basin and the I-CRI$_{SurfSoil}$ is shown. The seven shelf seas are abbreviated as CAA Canadian Arctic Archipelago, BFS Beaufort Sea, CS Chukchi Sea, ESS East Siberian Sea, LS Laptev Sea, KS Kara Sea; and BS Barents Sea. The maps of the Arctic Ocean in panels **a** and **b** are based on IBCAOv4[58,59].

Apart from the transport distance, the presence of permafrost or peat within the basin may have decisive impact on degradation rates under the different circum-Arctic freshwater regimes[48]. The two northernmost basins, the Canadian Arctic Archipelago and the East Siberian Sea, are underlain mostly by continuous permafrost (79 and 100%)[49] and show high I-CRI$_{SurfSoil}$ values, while the two southernmost basins (Beaufort and Kara seas) host only 43% and 17% of continuous permafrost (Fig. 4; Supplementary Fig. 1f)[49] and lower I-CRI$_{SurfSoil}$. These more southward-extending drainage basins are characterized by discontinuous permafrost and peatlands, which are known to provide deeper hydraulic conduits in the soil with longer reservoir times for transported terrOC[14], presumably leading to more efficient degradation[48]. Terrestrial OC transport through permafrost-free areas may thus contribute to degradation over long distances, and may contribute to the large differences of terrOC release along the circum-Arctic.

Comparing terrOC release with riverine $CO_2$ evasion provides yet another additional perspective on large-scale carbon cycling and terrOC remineralization in the circum-Arctic. The scale and spatial pattern of I-CRI are largely in line with previous estimates of inland $CO_2$ evasion from Arctic river catchments[50], and its relationship to the corresponding riverine export of total OC to the Arctic Ocean[31,51]. The atmospheric $CO_2$ evasion is 9-10 times higher than fluvial terrOC export in the large drainage basins of the Kara and Beaufort Seas (Fig. 4; Supplementary Fig. 1e)[31,50–52]. For the Ob River basin (Kara Sea) specifically, $CO_2$ emissions are up to nine times higher than the ocean export[53,54], with the highest $CO_2$ emissions in the more southerly, non-

continuous permafrost and peat-dominated part of the catchment. By contrast, the Laptev Sea catchment—to 80% underlain by continuous permafrost—reveals $CO_2$ evasion only twice that of the fluvial terrOC export. The basins of the Chukchi and East Siberian Seas, which are smaller and more northerly, also associate with notably less $CO_2$ evasion relative to the terrOC export ($CO_2$ evasion three and four times higher than export) than for the Kara and Beaufort Seas basins. This further supports the argument of stronger terrOC degradation in southerly and permafrost-free areas. In addition, this suggests the importance of efficient remineralization of released terrOC as part of the carbon cycle and constitutes a potentially significant source of greenhouse gas upon permafrost thaw and terrOC mobilization in the circum-Arctic.

## Synthesis

This study provides a large-scale receptor-based assessment of carbon releases from potentially climate-sensitive terrestrial deposits around the entire circum-Arctic. Dual-carbon isotope source apportionment and recipient fluxes in the circum-Arctic shelf seas reveal large differences in propensity for terrOC remobilization both from different source compartments and between the different Arctic Ocean drainage basins. The release of terrOC from the Eurasian-Arctic sector is about five times larger than from the American-Arctic sector. Most of the released terrOC originates from Surface Soil (61 ± 39%), with Ice Complex Deposit permafrost as the second largest source (30 ± 10%). The index metric of receptor-based terrOC fluxes to inland terrOC stocks (I-CRI) provides a perspective on the relative terrOC release

propensity along the entire circum-Arctic. The geographical patterns of the I-CRI suggest that one driver of terrOC release may be Arctic warming and the progressive permafrost thawing over the past half century, which broadly shows a spatial pattern similar to normalized terrOC release in the circum-Arctic. However, the normalized terrOC release varies by a factor of five between larger and smaller catchments, which also suggests efficient degradation of terrOC as part of long-distance freshwater transport and underway remineralization of terrOC to greenhouse gases. Taken together, this large-scale perspective on terrOC release patterns informs about the functioning of climate-relevant carbon remobilization across the circum-Arctic, and provides a benchmark for future investigations of terrOC release in the Arctic during climate change.

## Methods
### Organic carbon in surface sediments
Total OC concentrations as well as $\delta^{13}C$ and $\Delta^{14}C$ values of OC were taken from CASCADE[20,21] (https://bolin.su.se/data/cascade). Only data from surface sediments was considered in this study, which are here defined as the surface layer at the water-sediment interface and not deeper than 5 cm.

### Source apportionments of OC
This study employs a dual-isotope ($^{13}C$, $^{14}C$) mixing model[55] to distinguish fractions derived from the major OC sources (i.e., SurfSoil-OC, ICD-OC, Peat-OC, Petrogenic-C and marine phytoplankton) using the $\delta^{13}C$ and $\Delta^{14}C$ values of OC from 260 stations distributed across the Arctic Ocean (Fig. 2). This method simulates mixing of three end members using a Bayesian Markov chain Monte Carlo script[55], run in Matlab R2018a with 1,000,000 runs and a burn-in period of 10,000 runs per sample. The end members vary between the shelf seas depending on their occurrence in the catchment areas of the shelf seas. Of all end members, SurfSoil-OC, ICD-OC and peat OC were considered terrOC, while highly matured petrogenic OC released from, e.g., rock weathering or bituminous coal, abundant in the drainage areas of the Canadian Arctic Archipelago, Beaufort and Barents Seas[36,56] is unlikely to be a substantial contributor of greenhouse gases and was therefore excluded from the terrOC budget. For all shelf seas, mixing from three sources was thus assumed, consisting of two terrestrial and one marine end member, respectively. The end member definition was based on an extensive database, which has been utilized by a number of previous studies focusing on Siberian permafrost systems[5,17,57]. For this study, the database was substantially enhanced both geographically and numerically with additional and recently published data for previously unstudied regions. The end member definition for each shelf sea is described in detail in Supplementary Methods 1. A summary of the end members is given in Supplementary Table 4 and the full end member database is accessible in the Supplementary Data 1 spreadsheet.

### Cross-shelf transport correction
Previous studies have demonstrated that cross-shelf transport causes considerable aging of terrOC[26], which needs to be accounted for in the definition of the $^{14}C$ end member. Accordingly, our source apportionment script includes an approach described previously[26] to estimate OC aging with increasing cross-shelf distance for all terrOC end members. For transported SurfSoil-OC and peat OC, we assumed predominant transport by rivers and thus considered the distance between each $\Delta^{14}C$-location and the nearest river outlet (Supplementary Fig. 3a). For transported ICD-OC we used coast line compartments with active erosion[34] to measure the cross-shelf transport distance (Supplementary Fig. 3b). For both transport pathways the shortest distance between the sampling location and (i) the river outlet, or (ii) eroding coast line was considered, which is a conservative estimate and does not account for coast-parallel transport.

The effect of the cross-shelf transport correction was largest for the SurfSoil-OC end member. For instance, SurfSoil-OC transported to a location in the outer Laptev Sea 500 km away from the Lena river mouth would be affected by aging 2750 years[26], leading to a final $\Delta^{14}C$ value of transported SurfSoil-OC −433.9‰ (compared to the mean surface soil $\Delta^{14}C$ of −201.1‰). We also tested the sensitivity of the source apportionment and the OC accumulation rates to the cross-shelf transport correction. Source fractions calculated without considering cross-shelf transport revealed terrOC fractions $2 \pm 4\%$ larger (of the total OC; mean ± s.d.), while the relative uncertainties of the OC source fraction estimates were 11% higher. As this mostly affected samples at outer shelf locations with low mass accumulation rates there was no acknowledgeable effect to the mass accumulation budget. We further tested the effect of any post-depositional OC aging to the source apportionment. Based on $^{210}Pb_{xs}$ sedimentation rates in circum-Arctic shelf sediments ($0.21 \pm 0.22$ cm/yr average for this study; $n = 164$), and the corresponding $^{14}C$ depletion of marine OC over the 0–2 cm depth interval, we find resulting OC fractions to shift <1% (of the total OC) compared to source fractions for which the end member remained unchanged. Even under very low sedimentation rates (0.004 cm/yr), the (oldest) petrogenic OC fraction in Beaufort and Barents seas sediments is only around 1% smaller, which is much lower than the uncertainty estimates of this fraction. Hence, any post-depositional OC aging can be ruled out and will not significantly affect the source apportionment.

The results of the final source apportionments for all stations are provided in Supplementary Table 5.

### Estimation of mass accumulation rates (MAR)
To estimate the circum-Arctic accumulation of sediments we collected published MAR data based on $^{210}Pb$ dating of 152 shallow sediment cores. In addition, we performed gap-filling $^{210}Pb$ dating for 12 cores, of which ten were located in the Laptev and East Siberian seas, and two in the Kara Sea (Supplementary Table 7). A few studies provided information about sedimentation rates based on $^{210}Pb$ dating but no MAR was reported, in which cases a dry bulk density of 0.8 g cm$^{-3}$ was assumed to estimate the MAR[4]. Other cases reported OC fluxes only, which were converted into MAR using information about the OC concentration: MAR = OC flux (g m$^{-2}$ yr$^{-1}$) / OC (wt%) / 100. In total, our flux calculations are based on 164 cores dated using $^{210}Pb$ profiles (Supplementary Fig. 2; Supplementary Table 6). All MAR will also be included in the next version of CASCADE.

### Spatial interpolation and calculation of OC accumulation in circum-Arctic shelf seas
Calculated source fractions, total OC values from CASCADE and the MAR data were interpolated using Empirical Bayesian Kriging to an Arctic Ocean wide grid at 5 × 5 km resolution (EBK; ArcGIS, ESRI). For calculating fractional accumulation rates, OC fractions were interpolated and projected to the North Pole Lambert Azimuthal Equal Area coordinate system, which is a projection that is area conservative across the latitudes. The resulting interpolated layers were multiplied with interpolated total OC concentrations from CASCADE as well as with the gathered and interpolated MAR data.

### Calculation of carbon stocks in terrestrial deposits
To compare circum-Arctic accumulation of OC from terrestrial sources we calculated the stock of OC stored in surface soils (of both permafrost and non-permafrost soils), ICD and peatlands for each shelf sea watershed. Surface soils were defined as soils to a depth of 1 m. The calculation of the stock of OC in permafrost surface soils was facilitated by the NCSCD[22] using spatial OC densities for the active layer and permafrost to a depth of 100 cm. This dataset was amended by the Harmonized World Soil Database (HWSD)[23], which also includes non-permafrost soils at a vertical extent of 0–100 cm depth. To calculate

the OC stock in ICD permafrost in northeastern Siberia, Alaska and northwestern Canada, we calculated the ICD stock for each drainage basin using the fraction of the spatial coverage of ICD in the whole circum-Arctic and then allocated the total ICD-OC stock of 212 Pg C according to that distribution[3]. The OC stocks in circum-Arctic peatlands were calculated using the maps provided by Hugelius et al.[6]. It should be mentioned that peat overlaps with the surface soil 0–100 cm depth interval in areas of high peat coverage, such as for the Kara and Beaufort seas. All mapping was carried out in ArcGIS 10.6 (Esri, USA) and calculations were done in Matlab R2018a.

### Comparing accumulation of carbon in shelf seas with land-based carbon stocks I-CRI

Comparing the fractional OC accumulation in shelf sediments with the partitioning of the OC stored in the respective catchment in each of the circum-Arctic shelf seas informs about the tendencies of OC stores to be laterally released to the coastal sediments. The Integrated Carbon Release Index (I-CRI) per terrOC fraction ($f$, i.e., SurfSoil, ICD, Peat) represents the percentage (%) of the respective terrOC stock that is translocated to the shelf receptor per century (100 years). Hence, it is used to reflect the efficacy of terrOC release relative to the inland terrOC stock:

$$I-CRI_f[\% \text{ terrOC translocated per 100 yrs}] = \frac{MAR_f[\text{Tg yr}^{-1}]}{OC_f[\text{Pg}]} \times \frac{100\,\text{yrs}}{1000\,\text{Tg}} \times 100\% \tag{1}$$

Where $MAR_f$ is the mass of the fraction (of surface soil or ICD) accumulated to the sediments of each shelf sea, and $OC_f$ is the corresponding land-based OC inventory (of OC in surface soil or ICD). We also calculated the I-CRI for total terrOC using the sum of surface soil, ICD, and peat for MAR, and the total terrOC pool as OC inventory.

### Statistical analysis

Linear regression analysis was used to investigate the correlation between the I-CRI and the environmental characteristics of the drainage basins.

## Data availability

All data used and generated in this study are provided in the Supplementary Information. The data are also deposited in the Stockholm University Bolin Centre Database (https://bolin.su.se/data/martens-2022-arctic-terroc-1).

## Code availability

The Matlab code for the dual-isotope source apportionment will be made available under https://git.bolin.su.se/bolin/martens-2022-arctic-terroc.

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

## Acknowledgements

We thank the CASCADE collaboration partners and everyone else involved in the sampling and analysis that contributed to the first release of CASCADE. This study was supported by the European Research Council (ERC Advanced Grant CC-TOP 695331 to Ö.G.), the EU H2020-funded project Nunataryuk (Grant 773421 to Ö.G.), the Swedish Research Council (Grant 2017-01601 to Ö.G., Grant 2021-06670 to J.M.), and the Russian Science Foundation (grant 21-77-30001 to I.S.). Field campaigns to obtain gap-filling samples were supported by the Knut and Alice Wallenberg Foundation (KAW contract 2011.0027 to Ö.G.) as part of the SWERUS-C3 program and by the Russian Ministry of Science and higher Education (grant 075-15-2020-928 to HSE, grant 0211-2021-0010 to POI). This study was further supported by the Tomsk State University Development Programme (Priority-2030).

## Author contributions

The study was conceptualized by Ö.G., J.M., and B.W. The calculations, mapping, and drafting of the manuscript were carried out by J.M., under supervision and collaboration with Ö.G and B.W. J.M., B.W., I.S., O.D., and Ö.G. contributed to the data interpretation and the writing of the manuscript.

## Funding

## Competing interests

The authors declare no competing interests.
