## [Peer Review File · Nature Communications]

Circum-Arctic release of terrestrial carbon varies between regions and sourcesREVIEWER COMMENTS

Reviewer #1 (Remarks to the Author):

This manuscript uses circum-Arctic sediment OC data from the CASCADE data set, including stable and radiocarbon isotopes and Pb-210 accumulation data, to calculate the relative contribution of three types of terrestrial OC to carbon sequestration across Arctic sediments: surface soil OC, peat OC, and Ice Complex Deposit OC. The authors make some good conclusions about the relative sequestration rates of different sources of carbon, and how that relates to the amount of each type of OC in the terrestrial watersheds. However, there were some aspects of this discussion that were unclear to me that I think need to be revisited / rewritten before this manuscript is ready for publication.

Lines 52-53: "Lateral leakage of terrOC through aquatic conduits is a key component of the boundless carbon cycle in the often water-logged circum-Arctic." What exactly do you mean by "boundless" here? It seems like an odd word choice and the sentence might make more sense if it is taken out.

Figure 1: I think this figure could be cleaned up and made less busy. The orange, green, and brown color codes for ICD, surface soils, and peats are already in the legend, so I think the labels for those portions of the map can be removed (I realize the legend colors correspond to the pie charts, but the same colors are used for the map portion of the figure as well). I think there should be more detail in the figure caption; the solid black lines on the map appear to be watershed delineations while the solid red lines appear to delineate ICD, but again that's not clear. Try to utilize the legend more and have fewer words on the already busy map portion.

Figure 2 and I-CRI calculations: this calculation is unclear to me. The mass units in the I-CRI equation (line 324) do not cancel out ($Tg/Pg * 10$), so it's not a true fraction calculation and therefore is misleading. Because MAR is used, the ending units are in yr^{-1} , which means we're looking at an annual release, and from the way the text section is written (lines 121-134) it sounds like the authors are saying that "x fraction of y type of OC is lost from this watershed each year", which cannot be the case if $x > 1.0$ in many cases. I think this section needs to be rewritten so the description of exactly what the authors are calculating is more clear. Are these numbers meant to be percentages instead of fractions (x% of y type of OC is lost each year)? That would make more sense with the numbers reported.

Reviewer #2 (Remarks to the Author):

This manuscript presents new data on sediment accumulation rates (from fallout radionuclides) and bulk radiocarbon and stable isotope measurements on near surface sediments alongside a compilation (CASCADE) from the circum-Arctic shelf seas. It is tackling an important issue – the make-up of organic matter at the seafloor in the Arctic, and attempts to quantify source and rates of seafloor burial. The burial rates are higher than previous estimates by quite a margin. It then tries to assess what this then tells us about leaks of carbon from land (although it is actually dealing with burial, so this is somewhat confusing). The theme and dataset will surely be of interest, and some of these authors have contributed incredibly important work on this topic to date.

However, their mixing analysis is flawed, and not discussed with caveats presented. In my opinion there needs to be a much expanded discussion of the mixing analysis, with some independent variables tested to explore their plausibility (e.g. even if just testing outputs versus %OC would be helpful, and/or another variable such as OC/N or even the lipid extract insight from biomarkers).

Second, the logic of using of marine OC accumulation rates here to infer terrestrial losses isn't well explained, especially as a large part of the input by Arctic rivers is $<0.2-0.7$ microns in size (dissolved organic carbon) – and its not clear to me how this makes it to the outer shelf.

I expand on these major points below. I think that addressing this will lead to an important paper presenting a critical database.

1) Flaw in the mixing analysis: The mixing analysis uses radiocarbon and the stable carbon isotope composition to solve a three-component mixture (forward model). This three component mixture is not the same for different regions. This is nicely summarised in the data in Figure S2. First., this needs to be in the main text in my opinion – it is critical to the discussion – as does a rationale of why these end members are chosen and unique for each region.

However, there is a flaw in the analysis. The radiocarbon activity of surface marine sediments is not only set by the input composition of the organic matter (e.g. see papers by Griffith et al., 2010, GCA). It is also controlled by the residence time at the seafloor (set by sedimentation rate and mixed layer depth), and/or during lateral transport across vast shelf areas. In other words, decreases in D14C values can occur by aging of organic matter at the seafloor, either in situ, or during its transport in poorly constrained and understood shelf currents. While this is happening, selective degradation of compounds can lead to a shift in d13C values, leading to positive correlations between D14C and d13C that come about not due to mixing of end members with static compositions, but because of processes happening at the sea floor.

One way to check this is to look at the proportion of the most 14C-depleted end member in some of their mixing analysis. In the Barent Sea and Beaufort Sea, the 14C-depleted petrogenic end member is output from their model at 20-40% and >50% of the organic matter at the seafloor, respectively. For other regions, the ICD part (the 14C-depleted end member) is also quite high.

The authors have not checked whether this makes sense. For instance, what is the total %OC of these sediments? Its not given, but knowing the regions broadly, you could expect it is almost 2% in places in the BS and BFS. This would equate to 1% of rock-derived organic carbon in those marine sediments – which seems very high. We know some of this is weathered on land, and Galy's global compilation doesn't find many rivers with more than 0.4% rock organic carbon in the sediment load.

I expect that this apparent overestimate is likely due to aging of marine-derived organic matter at the seafloor, which then makes the whole analysis presented here very much over-simplified.

I'm not sure if this can all be addressed. But Figure S2 needs to be in the main text of this manuscript, and there needs to be expanded discussion on the organic matter sources, drawing in other evidence for source – first looking at %OC, but then also drawing in biomarker measurements that help to constrain source too.

2) DOC inputs from rivers vs OC burial on the seafloor: The Eurasian rivers deliver mostly DOC (see many reviews on this topic – Holmes and other works by the authors). The paper doesn't explain how this leads to accumulation of OC in the sediments at the seafloor – how does it get into the sediment mass?

A major thrust of the paper is the difference between the sediment accumulation at the seafloor and the estimated river fluxes (note we don't know the Canadian islands very well at all...). But there is no discussion of this key element – and thus any differences between these inputs and seafloor burial are over sold at the moment.

Reviewer #3 (Remarks to the Author):

Martens et al. examines how landscape heterogeneity relates to the proportion of terrestrially-derived OC in shelf sediments. They pose that an index - named the integrated carbon release (I-CRI), determined as the shelf terrOC accumulation fluxes with the corresponding catchment-specific terrOC stocks. The authors pull together an impressive and extensive list of end-members values for each of

the shelf sea regions, clearly presented in the supplemental tables. These likely present one of the largest potential errors of the papers approach, but it seems that the end members chosen for each and the values and ranges appear sensible and based upon the best existing data (esp. in S). It would however be highly advantageous for the Bayesian mixing model script used to run the analysis to be provided, rather than referencing another paper where a similar model was used. This would provide true replication of the results, and future updates as new sample measurements become available (maybe a link to an associate GitHub resource for instance?). This is particularly important if I-CRI is to become a comparable and useful metric of comparison between locations. The extensive CASCADE database is also a significant contribution, and well documented and linked to in the manuscript.

Overall, I think the authors have done a terrific job of synthesising such an extensive dataset and provide an interesting approach for examining future terr-OC release over Arctic regions. I support its publication with some minor modifications, largely involving inclusion of some supplemental data in the main report. I also would like to see Fig 1 be altered slightly, to reduce the strong red shelf color - which detracts from the rest of the image. A dark grey or other would be preferable. As above, I also think the code used for running the mixing model should be made freely available, as should the OC stocks you used (as noted in line 125 below).

I commend the authors however on really nice synthesis paper and presentation.

Additional Minor Comments.

Abstract.

19. I think the high I-CRI of ICD from the Beaufort sea/ relative to low I-CRI of terr-OC generally was also a noteworthy finding - and interesting to highlight as a potential comparison with the Siberian coastlines.

20. Unclear over what region these terrOC contribution estimates refer. I assume across the Arctic, but should be clarified.

24. This finding is interesting but the plots demonstrating these relationships are buried in the supplemental. As above, I would recommend moving some of these into the main text to support statements later in manuscript.

86. "this both supports to use"? Is this meant to read, this both supports the use of..

88. What are the relative number of points in the ESAS region then as compared to other Arctic sectors? Otherwise it just sounds like you are trying to push an agenda for more Siberian research.

92. Awkward grammar - "allows to compare?" Maybe reword i.e. "allow the propensity of xx to be compared".

102. Why are these combined for this region and in the plot. I'd like this explained. I now see having come back, due to the relative number of end members. This should be explained either in the text, or at least highlighted in the figure caption.

103. circum-Arctic not Artic.

104. should reference relationships in Supplemental or even better consider moving some of these into the main text to show relationships. Fig 1 is difficult to use to see relationships meaningfully.

107. "present in catchments of only three out of seven" but yet you show in Table S1 you show that it is present across 4? and use this in the Fig S1. (note Fig S1 would benefit from legend or additional text in figure caption - reminding reader of acronyms).

125. To provide transparency, you should provide a link here to supplemental material, or create some, showing the OC stocks you used for each basin and the sources of this material.

143. Yes, and the plot shown in supplemental is a really interesting finding that should be moved to main text (as should some others that show interesting patterns, i.e. warming trend and I-CRI (i.e. line 164)).

221. Demark? Demask doesn't make sense to me.

260. Again link to supplemental tables, or provide.

Throughout - be consistent with capitalization of circum vs Circum

Author responses to reviews and edits of *Nature Communications* manuscript “Circum-Arctic release of terrestrial carbon varies between regions and sources”

Ref: ms. no. NCOMMS-21-39219-T

Jannik Martens, Birgit Wild, Igor Semiletov, Oleg V. Dudarev, and Örjan Gustafsson

We gratefully thank the reviewers for constructive comments that have clearly contributed to improve the manuscript during revision. We are encouraged by the reviewers' recognition of the overall importance of this large-scale assessment of Arctic carbon cycling through statements such as “*the authors have done a terrific job of synthesizing such an extensive dataset and provide an interesting approach for examining future terr-OC release over Arctic regions*” (Reviewer 3).

Nevertheless, we also acknowledge issues raised by reviewers about the clarity of the manuscript and the level of detail in the Discussions and Methods sections. For instance, we recognize and have specifically acted on the comments raised by reviewer 2, which concern the robustness of the approach used in this study, the mixing model and the choice of the end members. Guided by these comments, we have thoroughly revised the entire manuscript and expanded the discussion by a more in-depth discussion about the source apportionment and the end members. We also now provide more background to the uncertainties that arise through the aging of organic matter and petrogenic carbon contributions.

All reviewer comments and our responses are listed below, organized such that each reviewer comment is shown first in *italics black font*, followed by our detailed response in normal blue tab-indented text. Our response refers to line numbers in the revised manuscript version.

Reviewer #1

This manuscript uses circum-Arctic sediment OC data from the CASCADE data set, including stable and radiocarbon isotopes and Pb-210 accumulation data, to calculate the relative contribution of three types of terrestrial OC to carbon sequestration across Arctic sediments: surface soil OC, peat OC, and Ice Complex Deposit OC. The authors make some good conclusions about the relative sequestration rates of different sources of carbon, and how that relates to the amount of each type of OC in the terrestrial watersheds. However, there were some aspects of this discussion that were unclear to me that I think need to be revisited / rewritten before this manuscript is ready for publication.

We are pleased that our manuscript is overall well-received and we are grateful for the input provided about some lack of clarity in the discussion, which has now been carefully revised.

Lines 52-53: “Lateral leakage of terrOC through aquatic conduits is a key component of the boundless carbon cycle in the often water-logged circum-Arctic.” What exactly do you mean by “boundless” here? It seems like an odd word choice and the sentence might make more sense if it is taken out.

The “*boundless carbon cycle*” is a commonly used concept introduced by Battin and co-workers (Battin et al., 2009). It means the dynamic changes to terrOC through varying input, exchanges and degradation along the land-ocean continuum. We agree with the reviewer that this term, without further explanation and discussion, may be confusing. Consequently, the word was removed here and elsewhere in the manuscript and replaced by “*carbon cycle*”.

Figure 1: I think this figure could be cleaned up and made less busy. The orange, green, and brown color codes for ICD, surface soils, and peats are already in the legend, so I think the labels for those portions of the map can be removed (I realize the legend colors correspond to the pie charts, but the

same colors are used for the map portion of the figure as well). I think there should be more detail in the figure caption; the solid black lines on the map appear to be watershed delineations while the solid red lines appear to delineate ICD, but again that's not clear. Try to utilize the legend more and have fewer words on the already busy map portion.

We appreciate the reviewer making this point, which helped us to improve the clarity of this figure. Accordingly, we removed the orange, green and brown compartment labels from the map, added more detail to the legend (incl. a description of the circle size representing terrOC fluxes), and improved the visibility of the shelf seas that are now more clearly distinguishable from the watershed delineations.

Figure 2 and I-CRI calculations: this calculation is unclear to me. The mass units in the I-CRI equation (line 324) do not cancel out ($Tg/Pg * 10$), so it's not a true fraction calculation and therefore is misleading. Because MAR is used, the ending units are in yr^{-1} , which means we're looking at an annual release, and from the way the text section is written (lines 121-134) it sounds like the authors are saying that “x fraction of y type of OC is lost from this watershed each year”, which cannot be the case if $x > 1.0$ in many cases. I think this section needs to be rewritten so the description of exactly what the authors are calculating is more clear. Are these numbers meant to be percentages instead of fractions (x% of y type of OC is lost each year)? That would make more sense with the numbers reported.

We agree with the reviewer that the calculation and the unit of the I-CRI were insufficiently explained. The reviewer is right that the index is defined to reflect the annual lateral translocation of terrOC per year ($Tg yr^{-1}$) relative to the inland stock (Pg). To clarify and also make the units more graspable, the I-CRI is now explained as percent (%) of terrOC that is translocated per century (100 years). We now describe this in detail in the main manuscript (line 159), with the unit of the I-CRI added at the first appearance (line 163):

“To this end, the Integrated Carbon Release Index (I-CRI) represents the percentage of the total terrOC, SurfSoil-OC, ICD-OC and Peat-OC stock in each drainage basin (Supplementary Table 2) (Panagos et al., 2012; Hugelius et al., 2013, 2020; Strauss et al., 2017) that accumulates in sediments over the course of 100 years (Supplementary Table 3).” (line 159-162)

Further, we revised the description of the I-CRI in the Methods (line 361) and added a definition of the unit to the equation of the I-CRI (line 366):

“The Integrated Carbon Release Index (I-CRI) per terrOC fraction (f , i.e. SurfSoil, ICD, Peat) represents the percentage (%) of the respective terrOC stock that is translocated to the shelf receptor per century (100 years). Hence, it is used to reflect the efficacy of terrOC release relative to the inland terrOC stock: “ (line 361-364)

$$I-CRI_f [\% \text{ terrOC exported per 100 yrs}] = \frac{MAR_f [Tg yr^{-1}]}{OC_f [Pg]} \times \frac{100 \text{ yrs}}{1000 Tg} \times 100\%$$

Reviewer #2

This manuscript presents new data on sediment accumulation rates (from fallout radionuclides) and bulk radiocarbon and stable isotope measurements on near surface sediments alongside a compilation (CASCADE) from the circum-Arctic shelf seas. It is tackling an important issue – the make-up of organic matter at the seafloor in the Arctic, and attempts to quantify source and rates of seafloor burial. The burial rates are higher than previous estimates by quite a margin. It then tries to assess what this then tells us about leaks of carbon from land (although it is actually dealing with burial, so this is somewhat

confusing). The theme and dataset will surely be of interest, and some of these authors have contributed incredibly important work on this topic to date.

However, their mixing analysis is flawed, and not discussed with caveats presented. In my opinion there needs to be a much expanded discussion of the mixing analysis, with some independent variables tested to explore their plausibility (e.g. even if just testing outputs versus %OC would be helpful, and/or another variable such as OC/N or even the lipid extract insight from biomarkers).

Second, the logic of using of marine OC accumulation rates here to infer terrestrial losses isn't well explained, especially as a large part of the input by Arctic rivers is <0.2-0.7 microns in size (dissolved organic carbon) – and its not clear to me how this makes it to the outer shelf.

I expand on these major points below. I think that addressing this will lead to an important paper presenting a critical database.

We appreciate the considerations raised by reviewer 2. However, we like to stress that the approach, to use shelf seas as receptors to study large-scale dynamics of terrestrial export, is well-established in earth science. Receptor-based approaches are even more frequent in atmospheric applications for e.g. estimating land-atmosphere fluxes. Shelf seas have repeatedly been demonstrated to serve well as receptors for understanding export patterns of land carbon, in the Arctic Ocean system and beyond. The many wide Arctic coastal shelf sea systems are used as accumulating receptors that reveal terrOC export patterns over large scales. For some shelf seas, the terrestrial imprint to the sediment OC is even larger than the marine carbon input. This is particularly true for the Arctic, e.g. along the East Siberian Arctic Shelf. Hence, an increasing number of studies use dual-isotope based source apportionment of OC (e.g., Vonk et al., 2012, *Nature*; Tesi et al., 2016, *Nat. Comm.*; Bröder et al., 2018, *Nat. Comm.*; Martens et al., 2020, *Sc. Adv.*), with continued effort to improve the isotope end members and to reduce uncertainties of OC source apportionment (e.g., Andersson et al., 2015; Vonk et al., 2015, 2017; Bröder et al., 2018).

The reviewer – characterizing our analysis as “*flawed*” – raises concerns about this well-established and carefully evaluated approach. We believe that some of this skepticism is rooted in misunderstanding of our method, which we admit was only described briefly in the introduction in the original manuscript, and in more detail in the text Supplementary Methods 1. We recognize that this approach may require additional elaboration and discussion in the main text. We have thus carefully revised the manuscript, and now provide detailed information about the mixing model and end member, and included also a more detailed reflection of the strengths and limitations of the approach. All raised points are addressed in further detail below.

1) Flaw in the mixing analysis: The mixing analysis uses radiocarbon and the stable carbon isotope composition to solve a three-component mixture (forward model). This three component mixture is not the same for different regions. This is nicely summarised in the data in Figure S2. First., this needs to be in the main text in my opinion – it is critical to the discussion – as does a rationale of why these end members are chosen and unique for each region.

We agree with the reviewer that the dual-isotope source apportionment and its underlying database of source-isotope fingerprints are important (see also the detailed description of end members in the Supplementary Methods). As described above, this dual-isotope source apportionment between different carbon pools have been applied in 10+ papers in the Arctic (with different source classes considered for different geographical regions) to date (e.g., Vonk et al., 2010, 2012; Tesi et al., 2014; Karlsson et al., 2016; Bröder et al., 2018, 2019; Grotheer et al., 2020) and also in other systems (e.g., Drenzek et al., 2009; Tao et al., 2015; Bao et al., 2018b; Hanke et al., 2019). To improve the overall clarity of the manuscript with regard to the approach and the selection of the end members, we followed the reviewer's recommendation and have now revised such that a short version of the source-fingerprint approach and data explained in Supplementary Methods 1 is now also presented in a new section in the main

discussion of the manuscript (starting in line 73). The detailed full description is still available in the Supplementary Information allowing anyone to reproduce the source apportionment calculations.

The following description was added to the discussion:

“Dual-isotope source apportionment in the circum-Arctic

The dual-isotope information ($\delta^{13}\text{C}/\Delta^{14}\text{C}$) provided by CASCADE allows us to distinguish between different terrOC sources and OC produced by marine phytoplankton. Given the heterogeneity of terrOC compartments around the circum-Arctic, this study employs different terrOC end members for each shelf sea (Supplementary Table 4, Supplementary Fig. 2), in each case based on large underlying end member collections of data on $\delta^{13}\text{C}$ and $\Delta^{14}\text{C}$ (Supplementary Text 2). Accordingly, OC from surface soils to a maximum depth of 100 cm was applied as end member in the entire circum-Arctic, with the exception of the Beaufort Sea where also deeper layers and peatlands were included due to overlapping isotopic compositions ($\delta^{13}\text{C}/\Delta^{14}\text{C}$) and limitation to three end members in the mixing model. Pleistocene age OC in ICD occurs in northeastern Siberia, Alaska and western Canada, and was hence considered as second terrOC end member in the Laptev, East Siberian, Chukchi and western Beaufort Sea shelves. Further, deep peat below 100 cm depth was considered as pre-aged terrOC end member for the Kara Sea, which has the world's largest peatland in its catchment. Petrogenic OC released from rock weathering was used as ^{14}C -depleted terrOC end member in the Canadian Arctic and the Barents Sea, to account for the significant release of petrogenic OC in these regions (Goñi et al., 2005; Drenzek et al., 2007).“ (lines 74-87)

However, there is a flaw in the analysis. The radiocarbon activity of surface marine sediments is not only set by the input composition of the organic matter (e.g. see papers by Griffith et al., 2010, GCA). It is also controlled by the residence time at the seafloor (set by sedimentation rate and mixed layer depth), and/or during lateral transport across vast shelf areas. In other words, decreases in $\Delta^{14}\text{C}$ values can occur by aging of organic matter at the seafloor, either in situ, or during its transport in poorly constrained and understood shelf currents. While this is happening, selective degradation of compounds can lead to a shift in $\delta^{13}\text{C}$ values, leading to positive correlations between $\Delta^{14}\text{C}$ and $\delta^{13}\text{C}$ that come about not due to mixing of end members with static compositions, but because of processes happening at the sea floor.

We understand that the reviewer thinks that we have missed to account for the ageing of terrOC during transport. This is a misunderstanding, which we take the blame for as we only explained this aspect in the Methods section. We agree with the reviewer that the ageing of OC may introduce uncertainty. We have published earlier on a key aspect that the reviewer brings up - the ageing of terrOC during cross-shelf transport of sediments (e.g., Bröder et al., 2018, *Nature Comm.*). This quantitative description of terrOC ageing has been used in several studies to account for ^{14}C changes during cross-shelf transport (e.g., Bröder et al., 2019; Martens et al., 2020, *Sc. Adv.*). The current study also accounts for this cross-shelf transport ageing for the terrOC end members, such that their source $\Delta^{14}\text{C}$ is not static but decreases with increasing offshore distance. Briefly, for OC released from surface soil and peat, we corrected for ageing of terrOC during its cross-shelf transport between the outlets of major Arctic rivers and each sampling location (Supplementary Fig. 4-A). For OC released from ICD, coastal erosion is the main translocation pathway (Vonk et al., 2012; Fritz et al., 2017), and we apply the offshore distance to sites of rapid coastal erosion (Supplementary Fig. 4-B) based on a large coastal database (Lantuit et al., 2012). The present study is the first that distinguishes between these different transport pathways when accounting for the ageing of terrOC during transport, which constitutes an advancement to earlier studies, yet builds on the great improvements on dual-isotope source apportionments of Arctic sediments that were accomplished by previous investigations (Bröder et al., 2018, 2019; Martens et al., 2020).

The reviewer also suggests a shift in $\delta^{13}\text{C}$ values may occur during aging. Our previous research on this topic, however, suggests extensive cross-shelf transport and aging to have no significant effect on the $\delta^{13}\text{C}$ composition of terrestrial compounds (Bröder et al., 2018). Another study revealed insignificant down-core trends of $\delta^{13}\text{C}$ of bulk OC in a near-coastal core in the East Siberian Sea, indicating little natural variability in $\delta^{13}\text{C}$ or $\Delta^{14}\text{C}$ before and after deposition in shelf sediments over a depositional period of >100 years (Bröder et al., 2016a).

Taken together, we agree that the potential changes in the isotopic composition of terrOC during transport (e.g., due to ageing) should be explained at greater detail. An explanation of the age-correction of terrOC during cross-shelf transport was hence added below the now extended discussion about the different terrOC sources (line 89):

“TerrOC end members were also corrected for aging during cross-shelf transport. Cross-shelf aging of terrOC was previously quantified by ^{14}C dating of terrestrial organic compounds (Bröder et al., 2018), which is here applied to correct for the transport distance of terrOC at each sampling location of the dataset (Methods, Cross-shelf transport correction). By contrast, the ^{13}C end member was assumed to stay constant as previous studies indicated continued terrestrial ^{13}C signatures despite major cross-shelf aging of terrOC (Bröder et al., 2018). Previous research also suggested aging of marine OC during cross-shelf transport (Mollenhauer et al., 2007; Bröder et al., 2016b; Bao et al., 2018a). However, the source location and the transport dynamics of marine OC are uncertain and the scale for marine OC aging in continental shelf seas is unknown, albeit likely much less than for terrOC due to both longer transports distances and lower recalcitrance.” (lines 89-97)

One way to check this is to look at the proportion of the most ^{14}C -depleted end member in some of their mixing analysis. In the Barent Sea and Beaufort Sea, the ^{14}C -depleted petrogenic end member is output from their model at 20-40% and >50% of the organic matter at the seafloor, respectively. For other regions, the ICD part (the ^{14}C -depleted end member) is also quite high.

We interpret this review comment to question whether a petrogenic contribution to total OC of 20-40% in the Barents Sea and >50% at some stations in the Beaufort Sea is reasonable. A range of studies have reported petrogenic OC in the Beaufort Sea, and have used different approaches to quantify its contribution to sediments. Based on a two end-member mixing model of young ($\Delta^{14}\text{C}$ 0‰) and fossil C sources ($\Delta^{14}\text{C}$ -1000‰), Goñi et al. (2005) estimated a contribution of petrogenic OC of 59–88% for shelf sediments and 59–71% for suspended sediments. Using a coupled ^{13}C - ^{14}C approach and biomarker-specific ^{14}C data, Drenzek et al. (2007) estimated that 40-50% of the OC in Beaufort Sea sediments are rock-derived. An alternative approach of source apportionment was followed by Hilton et al. (2015) and Vonk et al. (2015), where AI/OC ratios were used along with ^{14}C to estimate the age of a terrestrial, biospheric OC end member to be -501‰ (Vonk et al., 2015) and -514‰ (based on 5,800 years reported by Hilton et al., 2015). Based on their end members, the authors ultimately estimated 19±9% (Vonk et al., 2015) and 10-30% (Hilton et al., 2015) to be of petrogenic OC origin. We here applied a combined soil and peat end member (= terrestrial, biospheric OC) in the Beaufort Sea based on a much larger dataset than any of these earlier studies, including 191 ^{14}C -measurements of these source compartments in the Beaufort Sea catchment (Treat et al., 2016), which provides a reliable end member estimate of -378±201‰. We find our results (51±13% for the Beaufort Sea) to be well in line with the earlier independent reports of petrogenic OC fractions in the Beaufort Sea described above. We further note that the geographical distribution of the petrogenic OC fraction as estimated here is consistent with the location of previously identified petrogenic OC hotspots in other regions of the circum-Arctic (Yunker et al., 2011), such as the Barents Sea, for which petrogenic OC is a significant portion of the total OC in surface sediments (Kusch et al., 2021).

While we have excluded the petrogenic OC fraction from the terrOC budget, we find that this large OC source to Arctic shelf sediments should be acknowledged more centrally in the

manuscript. Therefore, the revised draft now contains a summary of the paragraph above, which not only discusses our results in the light of previous studies, but also acknowledges the limitations and uncertainties of the source apportionment. The new paragraph was inserted in line 143:

“In addition to terrOC from soils, permafrost and peat, petrogenic rocks release considerable amounts of OC in the catchments of the Canadian Arctic and off Svalbard. For the Canadian Arctic Archipelago and the Barents Sea, petrogenic OC accounted for $37\pm 26\%$ and $16\pm 11\%$ (mean \pm s.d.) of their total OC releases, respectively. For the Beaufort shelf, we find $51\pm 13\%$ of the OC in sediments to originate from petrogenic sources, which agrees with the large range (19-88%) suggested by previous studies specifically addressing this region (Goñi et al., 2005; Drenzek et al., 2007; Hilton et al., 2015; Vonk et al., 2015). Petrogenic OC in Beaufort Sea sediments can clearly be attributed to rock weathering and river export via suspended sediment material of the Mackenzie river and its tributaries (Hilton et al., 2015). Less certain, however, is whether petrogenic OC – likely to be highly recalcitrant – contributes to any notable emissions of CO₂ and thus affects the active carbon cycle, which is why petrogenic OC contributions were not considered terrOC release in the forthcoming part of the present study.” (line 143-153)

The authors have not checked whether this makes sense. For instance, what is the total %OC of these sediments? Its not given, but knowing the regions broadly, you could expect it is almost 2% in places in the BS and BFS. This would equate to 1% of rock-derived organic carbon in those marine sediments – which seems very high. We know some of this is weathered on land, and Galy's global compilation doesn't find many rivers with more than 0.4% rock organic carbon in the sediment load.

We agree with the reviewer that the Beaufort Sea receptor is indeed unlike most other receptors in the global ocean. We have in fact carefully validated our results with such published literature. As also espoused in the reply above, the Mackenzie drainage basin is well known to contain large amounts of exposed petroleum source rocks (Yunker et al., 2011; Hilton et al., 2015; Horan et al., 2019). As discussed above, many studies have demonstrated strong petroleum imprint on the total terrOC delivery in this specific Arctic region (e.g., Goñi et al., 2005, 2013; Drenzek et al., 2007; Yunker et al., 2011). The extensive dataset of OC concentrations in CASCADE yields that petrogenic OC concentrations in Beaufort Sea sediments are in the same range as reported for petrogenic OC concentrations of Mackenzie suspended material of 0.48 wt% (10-30% petrogenic OC of 1.6 wt%; Hilton et al., 2015). A statement about the suspended material from the Mackenzie river was added to the new paragraph about petrogenic OC:

“Petrogenic OC in Beaufort Sea sediments can clearly be attributed to rock weathering and river export via suspended sediment material of the Mackenzie river and its tributaries (Hilton et al., 2015).” (line 148-150)

I expect that this apparent overestimate is likely due to aging of marine-derived organic matter at the seafloor, which then makes the whole analysis presented here very much over-simplified.

We can follow the reviewer's concern and agree that also marine-derived OC may be subject to aging during transport processes. However, significant aging of marine OC likely occurs only at greater water depths, with little terrOC input and very low accumulation rates. Further, marine OC originates from phytoplankton and is ubiquitously sequestered in the ocean with transport routes being much less clear than for terrOC. Hence, it would also follow a different aging dynamic given generally higher primary production at the outer shelf. In addition, considering the dual-isotope data we find that ¹⁴C ages are low in regions where marine carbon dominates the organic carbon (e.g. Chukchi Sea, southern Barents Sea). These points together argue against aging of marine OC to lead to significant ¹⁴C aging along the shallow circum-Arctic shelf seas. Nonetheless, we like to acknowledge that marine OC aging may introduce

uncertainty, which should be clearly stated in the manuscript. We have now added a statement to the discussion that expands on the limitations and uncertainties of the dual-isotope approach (line 94).

“Previous research also suggested aging of marine OC during cross-shelf transport (Mollenhauer et al., 2007; Bröder et al., 2016b; Bao et al., 2018a). However, the source location and the transport dynamics of marine OC are uncertain and the scale for marine OC aging in continental shelf seas is unknown, albeit likely much less than for terrOC due to both longer transports distances and lower recalcitrance.” (lines 94-97)

I'm not sure if this can all be addressed. But Figure S2 needs to be in the main text of this manuscript, and there needs to be expanded discussion on the organic matter sources, drawing in other evidence for source – first looking at %OC, but then also drawing in biomarker measurements that help to constrain source too.

We are confident that we in the above sections have addressed all points raised by the reviewer, clarified misunderstandings, and demonstrated reasonableness of obtained results, and we have adjusted the manuscript accordingly. As detailed above, the discussion was amended by a new paragraph about the end member selection (lines 74-87) and a separate paragraph about C contributions from petrogenic sources (lines 143-153). We appreciate the reviewer suggesting to include also biomarker concentration in this study. While the first version of the CASCADE includes some biomarker observations, the data density is much less than for the isotopes. The limited type of biomarkers available are also difficult to use towards quantitatively addressing the importance of different sources. Hence, biomarkers are outside the scope of the current study and not needed for the core objectives.

Regarding Supplementary Fig. 1, we have broken out some of the information and included in Figure 3 in the main manuscript to now show the most important correlations more centrally. The remaining part of Supplementary Fig. 1, however, we suggest to keep in the Supplementary Information as this is a 9-panel figure with simple x-y scatter plots. At the editor's discretion, we will naturally agree to move these scatter plots into the main manuscript.

2) DOC inputs from rivers vs OC burial on the seafloor: The Eurasian rivers deliver mostly DOC (see many reviews on this topic – Holmes and other works by the authors). The paper doesn't explain how this leads to accumulation of OC in the sediments at the seafloor – how does it get into the sediment mass?

It is correct that most of the total OC export to the Arctic Ocean is in dissolved form (DOC). There are very few, if any, other shelf seas of the World Ocean that are relatively more impacted by terrOC input than the shelf seas of the semi-enclosed Arctic basin, with its massive input via both many rivers and by coastal erosion. The humic-rich shelf waters extend far north, affecting the optical density. While a substantial fraction of the terrestrial DOC may be transported long-distance off shelf, another fraction is very likely aggregating with increasing ionic strength in the marine receptor and then settling to the sediments below the unusually shallow water column. In fact, river DOC is well known to mix non-conservatively in Arctic shelf seas (e.g., Alling et al., 2010), indicating aggregation and settling when entering the Arctic Ocean. The sediment carbon on these – including the world's largest shelf sea systems – are dominated by terrestrial carbon (e.g., Vonk et al., 2012, Nature; Martens et al., 2021, Earth System Science Data). The text in the Introduction has been modified to elaborate some more on this aspect:

“By contrast, the seven Arctic shelf seas have the advantage to serve as natural integrators of terrOC release from the river drainage basins through the sequestration of riverine OC from dissolved and particular forms in their sediments upon aggregation and settling with increasing salinity (Alling et al., 2010). Moreover, shelf seas are recipient of OC from erosion of coastal permafrost deposits (ICD), which is suggested to be the dominating vector of terrOC release to

the extensive Laptev and East Siberian Seas (Vonk et al., 2012; Karlsson et al., 2016; Martens et al., 2020).” (line 53-58)

A major thrust of the paper is the difference between the sediment accumulation at the seafloor and the estimated river fluxes (note we don't know the Canadian islands very well at all...). But there is no discussion of this key element – and thus any differences between these inputs and seafloor burial are over sold at the moment.

We appreciate this comment, but do not agree with the reviewer's wordings. The core overarching objective of the paper is to explore large-scale spatial differences around the circum-Arctic in land-derived carbon fluxes to the corresponding shelf sea receptor systems. A direct comparison between the river OC fluxes and OC accumulation in shelf sediments is beyond the central scope of the manuscript for several reasons. Data on OC discharge and OC isotopes is only available for the largest rivers in the circum-Arctic, while also smaller river systems export significant amounts of terrOC. This leaves large uncertainties regarding the total terrOC export to the Arctic Ocean. Further, coastal erosion also adds substantial amounts of terrOC to the Arctic Ocean, which has been shown both by receptor and erosion-based estimates (Vonk et al., 2012; Wegner et al., 2015; Fritz et al., 2017; Terhaar et al., 2021). We therefore decided to not add a discussion about river fluxes to the revised version. Yet, we refer to the existing extended Supplementary Discussion 1 about previous results on terrOC fluxes (incl. river discharge) in the circum-Arctic (also referred to in line 121 in the main discussion).

Reviewer #3

Martens et al. examines how landscape heterogeneity relates to the proportion of terrestrially-derived OC in shelf sediments. They pose that an index - named the integrated carbon release (I-CRI), determined as the shelf terrOC accumulation fluxes with the corresponding catchment-specific terrOC stocks. The authors pull together an impressive and extensive list of end-members values for each of the shelf sea regions, clearly presented in the supplemental tables. These likely present one of the largest potential errors of the papers approach, but it seems that the end members chosen for each and the values and ranges appear sensible and based upon the best existing data (esp. in S). It would however be highly advantageous for the Bayesian mixing model script used to run the analysis to be provided, rather than referencing another paper where a similar model was used. This would provide true replication of the results, and future updates as new sample measurements become available (maybe a link to an associate GitHub resource for instance?). This is particularly important if I-CRI is to become a comparable and useful metric of comparison between locations. The extensive CASCADE database is also a significant contribution, and well documented and linked to in the manuscript.

We thank the reviewer for the supportive comments and agree that the exact Bayesian mixing model script should be published. The script is now prepared to be made accessible under <https://git.bolin.su.se/martens-2022> once the manuscript is accepted. We will also make the entire dataset as spreadsheets available under <https://bolin.su.se/data/martens-2022> upon acceptance of this manuscript. A statement about the code availability incl. the link was added to the manuscript (line 391-393).

Overall, I think the authors have done a terrific job of synthesising such an extensive dataset and provide an interesting approach for examining future terr-OC release over Arctic regions. I support its publication with some minor modifications, largely involving inclusion of some supplemental data in the main report. I also would like to see Fig 1 be altered slightly, to reduce the strong red shelf color - which detracts from the rest of the image. A dark grey or other would be preferable. As above, I also think the code used for running the mixing model should be made freely available, as should the OC stocks you used (as noted in line 125 below).

I commend the authors however on really nice synthesis paper and presentation.

We are pleased by the overall positive reception of our manuscript. For the revised version, we followed the reviewer's suggestion to include some of the supplementary data in the main manuscript. We also revised Figure 1 such that the red shelf sea borders are now removed and the shelf seas are now more clearly marked out. Further, the code of the mixing model is ready for publication upon acceptance of this paper as described above and in the revised manuscript.

All of the specific reviewer points are addressed below.

Additional Minor Comments.

Abstract.

19. *I think the high I-CRI of ICD from the Beaufort sea/ relative to low I-CRI of terr-OC generally was also a noteworthy finding - and interesting to highlight as a potential comparison with the Siberian coastlines.*

We agree with the reviewer that this is an interesting observation. However, the sentence in line 19 refers to the total terrOC accumulation and not the I-CRI. Given the limit of 150 words for the abstract, we are unable to expand on the regional differences of the I-CRI in the abstract.

20. *Unclear over what region these terrOC contribution estimates refer. I assume across the Arctic, but should be clarified.*

Thanks for pointing this out. The sentence was revised to “*Most of the circum-Arctic terrOC originates from ...*” (line 19)

24. *This finding is interesting but the plots demonstrating these relationships are buried in the supplemental. As above, I would recommend moving some of these into the main text to support statements later in manuscript.*

We appreciate this suggestion. Based on this reviewer input, we decided to integrate the regression plots for the I-CRI vs. temperature change and southward extent into Fig. 3. We agree that this significantly improves the comprehensibility of the discussion about the regional I-CRI differences.

86. *“this both supports to use”? Is this meant to read, this both supports the use of..*

Correct! The line was changed “*This both supports the use of...*” (line 111).

88. *What are the relative number of points in the ESAS region then as compared to other Arctic sectors? Otherwise it just sounds like you are trying to push an agenda for more Siberian research.*

This statement refers to the overall area of the Eurasian Arctic, which is in this study also found to hold the largest release of terrOC. We made this statement as the results clearly point at an urgent need for more research in this large area of the Arctic. We acknowledge, however, that this statement may be misunderstood and is unnecessary to conclude this paragraph. We thus decided to delete this sentence.

92. *Awkward grammar - “allows to compare?” Maybe reword i.e. “allow the propensity of xx to be compared”.*

The sentence was revised to “*The CASCADE facilitates also comparison of the propensity of...*” (line 115)

102. *Why are these combined for this region and in the plot. I'd like this explained. I now see having come back, due to the relative number of end members. This should be explained either in the text, or at least highlighted in the figure caption.*

We agree that this should be explained better in the Discussion. The end members of Surface Soil and Peat were combined in the Beaufort Sea as ^{14}C isotope endmembers of these terrestrial compartments overlap. This is now included in an additional part of the Discussion about the end member selection for all circum-Arctic shelf seas (line 78):

“Accordingly, OC from surface soils to a maximum depth of 100 cm was applied as end member in the entire circum-Arctic, with the exception of the Beaufort Sea where also deeper layers and peatlands were included due to overlapping isotopic compositions ($\delta^{13}\text{C}/\Delta^{14}\text{C}$) and limitation to three end members in the mixing model.” (line 78-81)

103. *circum-Arctic not Artic.*

Thanks for pointing out this typo. The word was corrected. (line 126)

104. *should reference relationships in Supplemental or even better consider moving some of these into the main text to show relationships. Fig 1 is difficult to use to see relationships meaningfully.*

We agree with the reviewer that it may not be fully clear how Figure 1 supports this important statement. This should be more clearly linked to the Supplement. There is, in fact, a significant positive relationship between the regional contributions to terrOC release and watershed-based % of the land-based terrOC stocks ($R^2=0.56$ and $p=0.05$), which is now included in Supplementary Fig. 1 and referred to in the main manuscript line 110 as follows:

“Moreover, the partitioning of terrOC release correlates with the relative distribution of the land-based terrOC stock in the circum-Arctic ($R^2=0.56$; $p=0.05$; Supplementary Fig. 1-I) (Hugelius et al., 2014, 2020; Strauss et al., 2017), with 78% of the SurfSoil-OC and 83% of the total peat OC being located within the Eurasian drainage basin (Panagos et al., 2012; Hugelius et al., 2013, 2020).” (line 108-111)

107. *“present in catchments of only three out of seven” but yet you show in Table S1 you show that it is present across 4? and use this in the Fig S1. (note Fig S1 would benefit from legend or additional text in figure caption - reminding reader of acronyms).*

We apologize for this mistake and the confusion. Its correct that ICD is present in four out of seven catchments. The sentence was revised accordingly and all abbreviations used in Supplementary Figure 1 were added to the figure caption.

125. *To provide transparency, you should provide a link here to supplemental material, or create some, showing the OC stocks you used for each basin and the sources of this material.*

A reference to the Supplementary Table 2 was added (line 161).

143. *Yes, and the plot shown in supplemental is a really interesting finding that should be moved to main text (as should some others that show interesting patterns, i.e. warming trend and I-CRI (i.e. line 164).*

We thank the reviewer for this suggestion, which has also been suggested by another reviewer. We are hesitant to include the full 9-panel X-Y scatter plot matrix in the main manuscript. However, for the two main discussion points (I-CRI vs. warming trends and I-CRI vs. fluvial degradation) we have included the regression plots to Fig. 3 to increase the visibility of these relationships.

221. *Demark? Demask doesn't make sense to me.*

We chose “demask” (as in demasking) to describe that the differences in terrOC release are masked if only the total terrOC fluxes are considered. By accounting for the terrOC stock in the catchment, the I-CRI reveals (or demasks) the propensity to terrOC release. We understand this message does not come across and the word was replaced by “*reveal*” (line 257).

260. *Again link to supplemental tables, or provide.*

The text in line 298 states: “*A summary of the end members is given in Supplementary Table 4 and the full end member database is accessible in the Supplementary Data 1 spreadsheet.*”

Throughout - be consistent with capitalization of circum vs Circum

Thanks for pointing this out. We are now consistently using “*circum-Arctic*” (except in the title, if a sentence starts with circum-Arctic or when CASCADE is spelled-out) throughout the manuscript.

References

- Alling, V., Sanchez-Garcia, L., Porcelli, D., Pugach, S., Vonk, J.E., Van Dongen, B., Mörth, C.M., Anderson, L.G., Sokolov, A., Andersson, P., Humborg, C., Semiletov, I., Gustafsson, Ö., 2010. Nonconservative behavior of dissolved organic carbon across the Laptev and East Siberian seas. *Global Biogeochem. Cycles* 24. <https://doi.org/10.1029/2010GB003834>
- Andersson, A., Deng, J., Du, K., Zheng, M., Yan, C., Sköld, M., Gustafsson, Ö., 2015. Regionally-varying combustion sources of the January 2013 severe haze events over eastern China. *Environ. Sci. Technol.* 49, 2038–2043. <https://doi.org/10.1021/es503855e>
- Bao, R., Uchida, M., Zhao, M., Haghypour, N., Montluçon, D., McNichol, A., Wacker, L., Hayes, J.M., Eglinton, T.I., 2018a. Organic Carbon Aging During Across-Shelf Transport. *Geophys. Res. Lett.* 45, 8425–8434. <https://doi.org/10.1029/2018GL078904>
- Bao, R., Voort, T.S. van der, Zhao, M., Guo, X., Montluçon, D.B., McIntyre, C., Eglinton, T.I., 2018b. Influence of Hydrodynamic Processes on the Fate of Sedimentary Organic Matter on Continental Margins. *Global Biogeochem. Cycles* 32, 1420–1432. <https://doi.org/10.1029/2018GB005921>
- Battin, T.J., Luysaert, S., Kaplan, L.A., Aufdenkampe, A.K., Richter, A., Tranvik, L.J., 2009. The boundless carbon cycle. *Nat. Geosci.* <https://doi.org/10.1038/ngeo618>
- Bröder, L., Andersson, A., Tesi, T., Semiletov, I., Gustafsson, Ö., 2019. Quantifying Degradative Loss of Terrigenous Organic Carbon in Surface Sediments Across the Laptev and East Siberian Sea. *Global Biogeochem. Cycles* 33, 85–99. <https://doi.org/10.1029/2018GB005967>
- Bröder, L., Tesi, T., Andersson, A., Eglinton, T.I., Semiletov, I.P., Dudarev, O. V, Roos, P., Gustafsson, Ö., 2016a. Historical records of organic matter supply and degradation status in the East Siberian Sea. *Org. Geochem.* 91, 16–30.
- Bröder, L., Tesi, T., Andersson, A., Semiletov, I., Gustafsson, Ö., 2018. Bounding the role of cross-shelf transport and degradation in land-ocean carbon transfer. *Nat. Commun.* 9, 806. <https://doi.org/10.1038/s41467-018-03192-1>
- Bröder, L., Tesi, T., Salvadó, J.A., Semiletov, I.P., Dudarev, O. V, Gustafsson, Ö., 2016b. Fate of terrigenous organic matter across the Laptev Sea from the mouth of the Lena River to the deep sea of the Arctic interior. *Biogeosciences* 13, 5003–5019. <https://doi.org/10.5194/bg-13-5003-2016>
- Drenzek, N.J., Huguen, K.A., Montluçon, D.B., Southon, J.R., dos Santos, G.M., Druffel, E.R.M., Giosan, L., Eglinton, T.I., 2009. A new look at old carbon in active margin sediments. *Geology* 37, 239–242. <https://doi.org/10.1130/G25351A.1>
- Drenzek, N.J., Montluçon, D.B., Yunker, M.B., Macdonald, R.W., Eglinton, T.I., 2007. Constraints on the origin of sedimentary organic carbon in the Beaufort Sea from coupled molecular ^{13}C and ^{14}C measurements. *Mar. Chem.* 103, 146–162. <https://doi.org/10.1016/j.marchem.2006.06.017>
- Fritz, M., Vonk, J.E., Lantuit, H., 2017. Collapsing Arctic coastlines. *Nat. Clim. Chang.* 7, 6–7. <https://doi.org/10.1038/nclimate3188>
- Goñi, M.A., O'Connor, A.E., Kuzyk, Z.Z., Yunker, M.B., Gobeil, C., Macdonald, R.W., 2013. Distribution and sources of organic matter in surface marine sediments across the North American Arctic margin. *J. Geophys. Res. Ocean.* 118, 4017–4035. <https://doi.org/10.1002/jgrc.20286>

- Goñi, M.A., Yunker, M.B., Macdonald, R.W., Eglinton, T.I., 2005. The supply and preservation of ancient and modern components of organic carbon in the Canadian Beaufort Shelf of the Arctic Ocean. *Mar. Chem.* 93, 53–73. <https://doi.org/10.1016/j.marchem.2004.08.001>
- Grotheer, H., Meyer, V., Riedel, T., Pfalz, G., Mathieu, L., Hefter, J., Gentz, T., Lantuit, H., Mollenhauer, G., Fritz, M., 2020. Burial and origin of permafrost derived carbon in the nearshore zone of the southern Canadian Beaufort Sea, in: *Geophysical Research Letters*. p. 2019GL085897. <https://doi.org/10.1029/2019GL085897>
- Hanke, U.M., Lima-Braun, A.L., Eglinton, T.I., Donnelly, J.P., Galy, V., Poussart, P., Hughen, K., McNichol, A.P., Xu, L., Reddy, C.M., 2019. Significance of perylene for source allocation of terrigenous organic matter in aquatic sediments. *Environ. Sci. Technol.* 53, 8244–8251. https://doi.org/10.1021/ACS.EST.9B02344/SUPPL_FILE/ES9B02344_SI_001.PDF
- Hilton, R.G., Galy, V., Gaillardet, J., Dellinger, M., Bryant, C., O'Regan, M., Gröcke, D.R., Coxall, H., Bouchez, J., Calmels, D., 2015. Erosion of organic carbon in the Arctic as a geological carbon dioxide sink. *Nature* 524, 84–87. <https://doi.org/10.1038/nature14653>
- Horan, K., Hilton, R.G., Dellinger, M., Tipper, E., Galy, V., Calmels, D., Selby, D., Gaillardet, J., Ottley, C.J., Parsons, D.R., Burton, K.W., 2019. Carbon dioxide emissions by rock organic carbon oxidation and the net geochemical carbon budget of the Mackenzie River Basin. *Am. J. Sci.* 319, 473–499. <https://doi.org/10.2475/06.2019.02>
- Hugelius, G., Loisel, J., Chadburn, S., Jackson, R.B., Jones, M., MacDonald, G., Marushchak, M., Olefeldt, D., Packalen, M., Siewert, M.B., Treat, C., Turetsky, M., Voigt, C., Yu, Z., 2020. Large stocks of peatland carbon and nitrogen are vulnerable to permafrost thaw. *Proc. Natl. Acad. Sci.* 201916387. <https://doi.org/10.1073/pnas.1916387117>
- Hugelius, G., Strauss, J., Zubrzycki, S., Harden, J.W., Schuur, E.A.G., Ping, C.-L., Schirrmeister, L., Grosse, G., Michaelson, G.J., Koven, C.D., O'Donnell, J.A., Elberling, B., Mishra, U., Camill, P., Yu, Z., Palmtag, J., Kuhry, P., 2014. Estimated stocks of circumpolar permafrost carbon with quantified uncertainty ranges and identified data gaps. *Biogeosciences* 11, 6573–6593. <https://doi.org/10.5194/bg-11-6573-2014>
- Hugelius, G., Tarnocai, C., Broll, G., Canadell, J.G., Kuhry, P., Swanson, D.K., 2013. The Northern Circumpolar Soil Carbon Database: spatially distributed datasets of soil coverage and soil carbon storage in the northern permafrost regions. *Earth Syst. Sci. Data* 5, 3–13.
- Karlsson, E., Gelting, J., Tesi, T., van Dongen, B., Andersson, A., Semiletov, I., Charkin, A., Dudarev, O., Gustafsson, Ö., 2016. Different sources and degradation state of dissolved, particulate, and sedimentary organic matter along the Eurasian Arctic coastal margin. *Global Biogeochem. Cycles* 30, 898–919. <https://doi.org/10.1002/2015GB005307>
- Kusch, S., Rethemeyer, J., Ransby, D., Mollenhauer, G., 2021. Permafrost Organic Carbon Turnover and Export Into a High-Arctic Fjord: A Case Study From Svalbard Using Compound-specific ¹⁴C Analysis. *J. Geophys. Res. Biogeosciences* 126, e2020JG006008. <https://doi.org/10.1029/2020JG006008>
- Lantuit, H., Overduin, P.P., Couture, N., Wetterich, S., Aré, F., Atkinson, D., Brown, J., Cherkashov, G., Drozdov, D., Donald Forbes, L., Graves-Gaylord, A., Grigoriev, M., Hubberten, H.W., Jordan, J., Jorgenson, T., Ødegård, R.S., Ogorodov, S., Pollard, W.H., Rachold, V., Sedenko, S., Solomon, S., Steenhuisen, F., Streletskaya, I., Vasiliev, A., 2012. The Arctic Coastal Dynamics Database: A New Classification Scheme and Statistics on Arctic Permafrost Coastlines. *Estuaries and Coasts* 35, 383–400. <https://doi.org/10.1007/s12237-010-9362-6>

- Martens, J., Romankevich, E., Semiletov, I., Wild, B., van Dongen, B., Vonk, J., Tesi, T., Shakhova, N., Dudarev, O. V., Kosmach, D., Vetrov, A., Lobkovsky, L., Belyaev, N., Macdonald, R.W., Pieńkowski, A.J., Eglinton, T.I., Haghypour, N., Dahle, S., Carroll, M.L., Åström, E.K.L., Grebmeier, J.M., Cooper, L.W., Possnert, G., Gustafsson, Ö., 2021. CASCADE – The Circum-Arctic Sediment Carbon Database. *Earth Syst. Sci. Data* 13, 2561–2572. <https://doi.org/10.5194/essd-13-2561-2021>
- Martens, J., Wild, B., Muschitiello, F., O'Regan, M., Jakobsson, M., Semiletov, I., Dudarev, O. V., Gustafsson, Ö., 2020. Remobilization of dormant carbon from Siberian-Arctic permafrost during three past warming events. *Sci. Adv.* 6, 6546–6562. <https://doi.org/10.1126/sciadv.abb6546>
- Mollenhauer, G., Inthorn, M., Vogt, T., Zabel, M., Sinninghe Damsté, J.S., Eglinton, T.I., 2007. Aging of marine organic matter during cross-shelf lateral transport in the Benguela upwelling system revealed by compound-specific radiocarbon dating. *Geochemistry, Geophys. Geosystems* 8, n/a-n/a. <https://doi.org/10.1029/2007GC001603>
- Panagos, P., Van Liedekerke, M., Jones, A., Montanarella, L., 2012. European Soil Data Centre: Response to European policy support and public data requirements. *Land use policy* 29, 329–338. <https://doi.org/10.1016/j.landusepol.2011.07.003>
- Strauss, J., Schirrmeyer, L., Grosse, G., Fortier, D., Hugelius, G., Knoblauch, C., Romanovsky, V., Schädel, C., von Deimling, T.S., Schuur, E.A.G., Shmelev, D., Ulrich, M., Veremeeva, A., 2017. Deep Yedoma permafrost: A synthesis of depositional characteristics and carbon vulnerability. *Earth-Science Rev.* 172, 75–86. <https://doi.org/10.1016/j.earscirev.2017.07.007>
- Tao, S., Eglinton, T.I., Montluçon, D.B., McIntyre, C., Zhao, M., 2015. Pre-aged soil organic carbon as a major component of the Yellow River suspended load: Regional significance and global relevance. *Earth and Planetary Science Letters*. <https://doi.org/10.1016/j.epsl.2015.01.004>
- Terhaar, J., Lauerwald, R., Regnier, P., Gruber, N., Bopp, L., 2021. Around one third of current Arctic Ocean primary production sustained by rivers and coastal erosion. *Nat. Commun.* 12, 1–10. <https://doi.org/10.1038/s41467-020-20470-z>
- Tesi, T., Muschitiello, F., Smittenberg, R.H., Jakobsson, M., Vonk, J.E., Hill, P., Andersson, A., Kirchner, N., Noormets, R., Dudarev, O. V., Semiletov, I.P., Gustafsson, Ö., 2016. Massive remobilization of permafrost carbon during post-glacial warming. *Nat. Commun.* 7, 13653. <https://doi.org/10.1038/ncomms13653>
- Tesi, T., Semiletov, I., Hugelius, G., Dudarev, O., Kuhry, P., Gustafsson, Ö., 2014. Composition and fate of terrigenous organic matter along the Arctic land–ocean continuum in East Siberia: Insights from biomarkers and carbon isotopes. *Geochim. Cosmochim. Acta* 133, 235–256. <https://doi.org/10.1016/j.gca.2014.02.045>
- Treat, C.C., Jones, M.C., Camill, P., Gallego-Sala, A., Garneau, M., Harden, J.W., Hugelius, G., Klein, E.S., Kokfelt, U., Kuhry, P., Loisel, J., Mathijssen, P.J.H., O'Donnell, J.A., Oksanen, P.O., Ronkainen, T.M., Sannel, A.B.K., Talbot, J., Tarnocai, C., Väliranta, M., 2016. Effects of permafrost aggradation on peat properties as determined from a pan-Arctic synthesis of plant macrofossils. *J. Geophys. Res. Biogeosciences* 121, 78–94. <https://doi.org/10.1002/2015JG003061>
- Vonk, J., Sánchez-García, L., van Dongen, B.E., Alling, V., Kosmach, D., Charkin, A., Semiletov, I.P., Dudarev, O. V., Shakhova, N., Roos, P., Eglinton, T.I., Andersson, A., Gustafsson, Ö., Gustafsson, O., 2012. Activation of old carbon by erosion of coastal and subsea permafrost in Arctic Siberia. *Nature* 489, 137–140. <https://doi.org/10.1038/nature11392>
- Vonk, J.E., Giosan, L., Blusztajn, J., Montluçon, D., Graf Pannatier, E., McIntyre, C., Wacker, L., Macdonald, R.W., Yunker, M.B., Eglinton, T.I., 2015. Spatial variations in geochemical characteristics

of the modern Mackenzie Delta sedimentary system. *Geochim. Cosmochim. Acta* 171, 100–120. <https://doi.org/10.1016/j.gca.2015.08.005>

Vonk, J.E., Sánchez-García, L., Semiletov, I., Dudarev, O., Eglinton, T., Andersson, A., Gustafsson, Ö., 2010. Molecular and radiocarbon constraints on sources and degradation of terrestrial organic carbon along the Kolyma paleoriver transect, East Siberian Sea. *Biogeosciences* 7, 3153–3166.

Vonk, J.E., Tesi, T., Bröder, L., Holmstrand, H., Hugelius, G., Andersson, A., Dudarev, O., Semiletov, I., Gustafsson, Ö., 2017. Distinguishing between old and modern permafrost sources in the northeast Siberian land–shelf system with compound-specific $\delta^2\text{H}$ analysis. *Cryosph.* 11, 1879–1895. <https://doi.org/10.5194/tc-11-1879-2017>

Wegner, C., Bennett, K.E., de Vernal, A., Forwick, M., Fritz, M., Heikkila, M., Lacka, M., Lantuit, H., Laska, M., Moskalik, M., O'Regan, M., Pawlowska, J., Prominska, A., Rachold, V., Vonk, J.E., Werner, K., 2015. Variability in transport of terrigenous material on the shelves and the deep Arctic Ocean during the Holocene. *Polar Res.* 34, 1–19. <https://doi.org/10.3402/polar.v34.24964>

Yunker, M.B., Macdonald, R.W., Snowdon, L.R., Fowler, B.R., 2011. Alkane and PAH biomarkers as tracers of terrigenous organic carbon in Arctic Ocean sediments. *Org. Geochem.* 42, 1109–1146. <https://doi.org/10.1016/j.orggeochem.2011.06.007>

REVIEWER COMMENTS

Reviewer #1 (Remarks to the Author):

I think this manuscript is much improved, and the authors have addressed all comments well. I recommend for publication.

Reviewer #2 (Remarks to the Author):

I stand by my opinion that a forward model with two compositions and three end members that are spatially variable and themselves hold a lot of variability (e.g. soils in D14C) could be improved. Testing the outputs of the model against another measured parameter would be ideal. e.g. %OC (which the authors did not clearly do in their reply) or %N or C/N perhaps. I know we're not able to do this routinely, and I am of course aware of the studies that have used the approach used here, it doesn't mean it is the right one, and we shouldn't aim to validate the outputs and think carefully about the caveats of the approach.

And I do agree with the other reviewers on how useful this database is, and that the theme is important. It's just that the analysis is overly confident in my opinion.

I still think that more should be explained about the analysis in the main text, with supplementary Figure 2 very important here. Given the outputs are correlated with independent variables (on climate), it's very important to understand the potential caveats in the approach.

1) on the potential for OM aging and $\delta^{14}\text{C}$ -shifts not related to end member mixing:

It was very helpful to have the clarification that cross shelf transport was considered, thank you. However, can the authors explain the impact this had on their mixing analysis. Presumably, it shifts points in the Figure S2 $\delta^{14}\text{C}$ vs $\delta^{13}\text{C}$ space? By how much?

the approach does not consider the residence time of OM at the seafloor. When sampled, some of the core top sediments (1-2 cm) may have been residing there for thousands of years. I wasn't clear that this was assessed.

Looking at Figure S2, this could explain a large part of the $\delta^{14}\text{C}$ depletion in the CS, CA, and Barents Sea - i.e. that much more of the % was derived originally from a marine composition that has been partially stabilised and buried at the seafloor. It would mean that the non-marine "residue" from the mixing analysis is highly uncertain, and make the basin wide comparison attempted here very difficult to ground.

Author response to reviews and resulting edits of *Nature Communications* manuscript “Circum-Arctic release of terrestrial carbon varies between regions and sources”

Ref: ms. no. NCOMMS-21-39219A

Jannik Martens, Birgit Wild, Igor Semiletov, Oleg V. Dudarev, and Örjan Gustafsson

We gratefully thank the reviewers for thoroughly examining our revised manuscript; this has contributed to further sharpen the manuscript. We are delighted that reviewer 1 is now fully backing publication. We are also glad that reviewer 2 acknowledges that most issues are satisfactorily resolved through R1 and we believe that the few remaining issues raised by reviewer 2 are now addressed in this short R2 stage, further improving both quality and clarity of this paper.

Below, we list the few remaining comments by reviewer 2 (*italics black font*) followed by our detailed responses (in blue text). We first address the general comments raised in the introductory paragraph of review 2 by dividing these up into its three parts (general comments 1-3), and then address the one specific review comment in its three parts (specific comments 1a-c). Our responses refer to line numbers of the re-revised manuscript version.

Reviewer #1

I think this manuscript is much improved, and the authors have addressed all comments well. I recommend for publication.

We gratefully thank reviewer 1 for reading and re-considering our revised manuscript. We are pleased about the clear endorsement for publication.

Reviewer #2

General comment 1: I stand by my opinion that a forward model with two compositions and three end members that are spatially variable and themselves hold a lot of variability (e.g. soils in DI4C) could be improved. Testing the outputs of the model against another measured parameter would be ideal. e.g. %OC (which the authors did not clearly do in their reply) or %N or C/N perhaps. I know we're not able to do this routinely, and I am of course aware of the studies that have used the approached used here, it doesn't mean it is the right one, and we shouldn't aim to validate the outputs and think carefully about the caveats of the approach.

We naturally agree with the spirit of the reviewer's comment that it is always important to clarify weaknesses and uncertainties of any approach; we are convinced the present manuscript does this, yet we seek to further improve and clarify this part. However, it is not possible to follow the suggestion to test the model output against %OC, %N or C/N because the dual-isotope source apportionment does not have any of the three parameters as output. The dual-isotope mass balance deconvolutes the relative contribution of different OC sources to the measured OC concentrations in the receptor. Given the lack of any further suggestion from the reviewer how this should be done, it seems likely that this suggestion builds on a misunderstanding of our method as the approach suggested by the reviewer is simply not feasible. Below, we again outline the current approach and describe how uncertainty is estimated therein. To alleviate any misunderstandings, we have also scrutinized and improved the clarity of the description of the dual-isotope approach throughout the manuscript.

The isotope-mass balance approach to source apportionment in the current manuscript builds on earlier studies, yet the current manuscript has a wider scope in scientific application. It is fundamentally an inverse receptor-based approach building on observational constraints. The approach benefits from the sharp source-diagnostic dual-isotope fingerprint of OC in shelf sediments, which is compared with substantial databases of the $\Delta^{14}\text{C}/\delta^{13}\text{C}$ fingerprints of different OC sources. Further, the method employs Bayesian statistics within a Markov Chain Monte Carlo simulation to realistically constrain uncertainties of the OC fraction estimates.

The natural variability of the end member compartments is accounted for by using large collections of $\Delta^{14}\text{C}$ and $\delta^{13}\text{C}$ measurements of each specific source compartment (explained in lines 300-305 and Supplementary Text 2). The end member variability is statistically defined as mean \pm standard deviation (s.d.) and contributes to the overall estimated uncertainty of the source apportionment (Supplementary Table 5). The present study builds on previous work that gathered large end-member data collections with hundreds of ^{13}C and ^{14}C measurements (e.g., Vonk et al., 2012; Karlsson et al., 2016; Wild et al., 2019; Martens et al., 2020). To further improve the end member representation and spatial coverage of observations, we further updated and refined this end member data collection as part of this study.

As espoused above, it is unclear to us how the reviewer's suggestion of comparing with measurements of %OC would constitute an advance towards testing this statistically-robust dual-isotope approach, which provides clear distinction between well-defined terrestrial and marine sources and comprehensively constrains the uncertainty.

However, we recognize that the manuscript may benefit from further description of the uncertainty of the end members and that of the source fractions. We thus added an explanation about the end member variability to the respective paragraph in the discussion: "*Each end member is described by the mean \pm standard deviation (s.d.) of the underlying data collection, which represents the natural variability of the different source compartments and is mirrored in the uncertainty of the resulting source fractions.*" (line 88-90)

We have also included a statement later into the discussion to remind the reader about the uncertainty: "*Although the dual-isotope receptor approach holds uncertainty, these results support the use of shelf sediments to deduce large-scale features of terrOC release and underscore the importance of the Eurasian-Arctic system for terrOC vulnerability in the Arctic.*" (line 115-118)

General comment 2: *And I do agree with the other reviewers on how useful this database is, and that the theme is important. Its just that the analysis is overly confident in my opinion.*

We appreciate that the reviewer in this round describes our database as useful and the theme as important. We hope that the revised version of this manuscript addresses all remaining concerns.

General comment 3: *I still think that more should be explained about the analysis in the main text, with supplementary Figure 2 very important here. Given the outputs are correlated with independent variables (on climate), its very important to understand the potential caveats in the approach.*

We agree with the reviewer that analysis methods and uncertainties should be discussed and explained clearly. In addition to all such information that is already in the manuscript, we have now also added more information about the end member uncertainty to the manuscript as described above (line 88-90 and 115-116).

Given the results and implications of the I-CRI (the Integrated Carbon Release Index), we have also added a further clarifying statement about the uncertainty of the I-CRI (line 167): "*To this end, the Integrated Carbon Release Index (I-CRI) represents the percentage of the total terrOC, SurfSoil-OC, ICD-OC and Peat-OC stock in each drainage basin (Supplementary Table 2)^{3,6,22,23} that accumulates in sediments over the course of 100 years (Supplementary Table 3), where the estimate of the total uncertainty (s.d.) includes the spatial variability of OC accumulation fluxes, as well as the uncertainties of the source apportionment and that of the land-based terrOC stock estimates.*" (line 164-169)

Specific review comments:

1) on the potential for OM aging and 14C-shifts not related to end member mixing:

a) It was very helpful to have the clarification that cross shelf transport was considered, thank you. However, can the authors explain the impact this had on their mixing analysis. Presumably, it shifts points in the Figure S2 D14C vs d13C space? By how much?

We are pleased that our clarifications are well-received. It is correct that the cross-shelf transport correction shifts the $\Delta^{14}\text{C}/\delta^{13}\text{C}$ points in Figure S2, i.e. towards more negative $\Delta^{14}\text{C}$ values with increasing transport distance. As described in the manuscript (line 319-322), the surface soil end member with initially (near-coastal) -201‰ is -434‰ at a 500 km offshore distance. We also note that, this approach was established and analytically tested using ^{14}C measurements of specific terrestrial biomarkers along a cross-shelf transect in the Laptev Sea (Bröder et al., 2018)(line 308-317).

It is correct and also desired that the $\Delta^{14}\text{C}$ correction affects the results. We carefully tested the influence of terrOC aging to the source apportionment during data analysis and found it to cause only small differences among the OC fractions across the dataset, with terrOC fractions (of the total OC) being $2\pm 4\%$ (mean \pm s.d.) smaller when the correction is applied. Additionally, the correction reduced the relative uncertainty of the OC source fraction estimates by about 11%. However, we like to stress that despite these small differences (and improvements), the use of the transport correction has no effect to the main interpretations and conclusions of this study.

Please note that also the results of the sensitivity test are described in the Methods between lines 322-327. We have now slightly revised the writing to also inform that relative uncertainties are up to 11% higher without the transport correction: “Source fractions calculated without considering cross-shelf transport revealed terrOC fractions $2 \pm 4\%$ larger (of the total OC; mean \pm s.d.), while the relative uncertainties of the OC source fraction estimates were 11% higher.” (line 323-326)

b) the approach does not consider the residence time of OM at the seafloor. When sampled, some of the core top sediments (1-2 cm) may have been residing there for thousands of years. I wasn't clear that this was assessed.

Aging of the scale indicated by the reviewer is highly unlikely in the studied system. Aging of the residual marine OC at the sea floor is much shorter in our study area due to the overall high sedimentation rates (dataset average $0.21 \pm 0.22 \text{ cm yr}^{-1}$; $n=164$) in Arctic shelf seas, shown by this and numerous previous studies using $^{210}\text{Pb}_{\text{xs}}$ and ^{14}C dating. Based on the $^{210}\text{Pb}_{\text{xs}}$ sedimentation rates, we calculate that it takes 10 ± 11 years to accumulate a layer of 1 cm at the sediment surface, while a minimum of 70 years would be needed to resolve any ^{14}C depletion on the $\Delta^{14}\text{C}$ scale. This is also shown by “modern” (1950-today) ^{14}C activities of marine carbonate fossils retrieved from the top 3-4 cm of Arctic surface sediments, indicating that depositional ages were negligible on the ^{14}C timescale (Bröder et al., 2018).

c) Looking at Figure S2, this could explain a large part of the $\Delta^{14}\text{C}$ depletion in the CS, CA, and Barents Sea - i.e. that much more of the % was derived originally from a marine composition that has been partially stabilized and buried at the seafloor. It would mean that the non-marine “residue” from the mixing analysis is highly uncertain, and make the basin wide comparison attempted here very difficult to ground.

Based on our explanations and discussions above, we can exclude that OC residence in the surface sediment layer may explain any significant part of the observed ^{14}C depletion in the surface sediments of the Chukchi Sea, the Canadian Archipelago, the Barents Sea or any other of the studied circum-Arctic shelf sea systems.

We now added a line to the manuscript to clarify why any effect of in-situ OC aging to the surface sediment $\Delta^{14}\text{C}$ is highly unlikely: “We further note that based on overall high sedimentation rates in circum-Arctic shelf sediments ($0.21 \pm 0.22 \text{ cm/yr}$ average for this study; $n= 164$), any post-depositional OC aging can be excluded to significantly affect the source apportionment.” (line 327-330)

References

- Bröder, L., Tesi, T., Andersson, A., Semiletov, I., Gustafsson, Ö., 2018. Bounding the role of cross-shelf transport and degradation in land-ocean carbon transfer. *Nat. Commun.* 9, 806. <https://doi.org/10.1038/s41467-018-03192-1>
- Karlsson, E., Gelting, J., Tesi, T., van Dongen, B., Andersson, A., Semiletov, I., Charkin, A., Dudarev, O., Gustafsson, Ö., 2016. Different sources and degradation state of dissolved, particulate, and sedimentary organic matter along the Eurasian Arctic coastal margin. *Global Biogeochem. Cycles* 30, 898–919. <https://doi.org/10.1002/2015GB005307>
- Martens, J., Wild, B., Muschitiello, F., O'Regan, M., Jakobsson, M., Semiletov, I., Dudarev, O. V., Gustafsson, Ö., 2020. Remobilization of dormant carbon from Siberian-Arctic permafrost during three past warming events. *Sci. Adv.* 6, 6546–6562. <https://doi.org/10.1126/sciadv.abb6546>
- Vonk, J., Sánchez-García, L., van Dongen, B.E., Alling, V., Kosmach, D., Charkin, A., Semiletov, I.P., Dudarev, O. V., Shakhova, N., Roos, P., Eglinton, T.I., Andersson, A., Gustafsson, Ö., Gustafsson, O., 2012. Activation of old carbon by erosion of coastal and subsea permafrost in Arctic Siberia. *Nature* 489, 137–140. <https://doi.org/10.1038/nature11392>
- Wild, B., Andersson, A., Bröder, L., Vonk, J., Hugelius, G., McClelland, J.W., Song, W., Raymond, P.A., Gustafsson, Ö., 2019. Rivers across the Siberian Arctic unearth the patterns of carbon release from thawing permafrost. *Proc. Natl. Acad. Sci. U. S. A.* 201811797. <https://doi.org/10.1073/pnas.1811797116>

REVIEWER COMMENTS

Reviewer #2 (Remarks to the Author):

I include some more detail of the “test” I was proposing of their mixing analysis below, in addition to clarifying an earlier request to move the Supplementary Figure 2 into the main text of the article.

General Comment 1

Author reply - “We naturally agree with the spirit of the reviewer's comment that it is always important to clarify weaknesses and uncertainties of any approach; we are convinced the present manuscript does this, yet we seek to further improve and clarify this part. However, it is not possible to follow the suggestion to test the model output against %OC, %N or C/N because the dual-isotope source apportionment does not have any of the three parameters as output.”

Response - This is precisely my point. Because the analysis does not output these, they can be used independently to explore how the model is behaving. The key question is are the fractions of carbon from each end members re-producing sensible bulk elemental proxies? (note you could also imagine an analysis which uses d15N as test). So, in an ideal world, you would have a set of data which is not being used in the mixing analysis, from which the mixing outputs could be explored.

Author reply - “Given the lack of any further suggestion from the reviewer how this should be done, it seems likely that this suggestion builds on a misunderstanding of our method as the approach suggested by the reviewer is simply not feasible.”

Response - My apologies for not expanding. The underlying maths is straightforward and similar to those underpinning the approach applied in the paper, so I thought it would clear what I was getting at. Thinking about the organic carbon concentration example, which I imagine you should have the data in your compilation (although it is not in the data files), then we can write that f_a – which is the fraction of carbon from end member a (an output of the dual isotope mixing analysis) – is equal to:

$$f_a = [\text{OC}]_a / [\text{OC}]_{\text{tot}}$$

where $[\text{OC}]_{\text{tot}}$ is the total organic carbon concentration in the sample being unmixed, while $[\text{OC}]_a$ is the organic carbon concentration in the sample mass brought by end member a (note I have skipped a step in the algebra involving the conversation of mass of carbon to concentration of carbon).

So, if we output f_a , we can quickly calculate a predicted $[\text{OC}]_a$, just knowing the organic carbon concentration in the sample.

For this study, a logical component to look at as a “test” would be the petrogenic component, as that $[\text{OC}]_{\text{petro}}$ will reflect a pre-mixture of organic matter and mineral matrix, and rock organic carbon concentrations are measured a lot for other reasons. One can calculate the expected $[\text{OC}]_{\text{petro}}$, and see if it makes sense. If its high, perhaps the aged OM end member is more important than the mixing model suggests based on the input end member. Or, perhaps the aging during transport or residence at the seafloor has been underestimated.

As an example, one could look at the BFS samples listed in Supplementary Table 5. Just picking a sample here to show what I mean, PG2318-1 (where the $[\text{OC}]_{\text{tot}}$ is reported in a master thesis online, Riedel, 2017, as $[\text{OC}]_{\text{tot}} = 1.3\%$) has a $f_{\text{petro}} = 0.52$. This suggests a $[\text{OC}]_{\text{petro}} = 0.65\%$. For PG2313-1, $[\text{OC}]_{\text{tot}} = 2.1\%$ and f_{petro} is reported here = 0.60. This suggests a $[\text{OC}]_{\text{petro}} = 1.26\%$. These do seem high, the second one especially, when they are compared to the rivers in the region (refs 35 and 36 in the manuscript). One could plot these predicted $[\text{OC}]_{\text{petro}}$ for different sites, and see if the numbers makes sense.

Overall, I hope this quick analysis is easy to follow for the authors and clarifies what I meant in my last 2 reviews. It suggests a petrogenic component may be being overestimated and an old soil organic matter component underestimated. This could mean the Beaufort Sea circles and %surface in the figures have more uncertainty?

I haven't done this wider in the dataset, but the petrogenic end member check could also be used for the Barent Sea.

Response to "General Comment 3"

The small clarifications in the main text are welcome.

However, please put Figure S2 in the main text... For a paper like this in Nat comms, most readers won't understand fully what data has been used to produce the figures or the "index" here. For me, that is a problem, yet the fix is really straightforward – Supplementary Figure 2 becomes a main text figure. These are the raw measurements which we have confidence in. The figure also shows the compilations of the end members very nicely. Finally, it shows the decision making by the authors that have selected different end members for each basin. It really is an important figure to have in the main paper, not hidden away.

Specific review comments 1b & c (on potential OM aging):

A quick look on a reference search finds reported Barent Sea sedimentation rates from 0.004 to 0.21 cm/yr, average 0.06 cm/yr (compiled in the publication Faust et al., 2020, Philosophical Transactions of the Royal Society A). Looking at the figures in that publication, a number of sites away from coastal settings have sedimentation rates <0.01 cm/yr. So in that context, the upper 2 cm could have shifts in D14C of approximately ~30 permil due to aging. If is the extreme of that range (0.004 cm/yr), it would be a shift of about ~70 permil. Is this worth noting? 0.21+-0.22 is quite a range, and clearly the lower sedimentation rate sites could be impacted by in-situ aging?

Author response to reviews and resulting edits of *Nature Communications* manuscript “Circum-Arctic release of terrestrial carbon varies between regions and sources”

Ref: ms. no. NCOMMS-21-39219B

Jannik Martens, Birgit Wild, Igor Semiletov, Oleg V. Dudarev, and Örjan Gustafsson

We gratefully thank the reviewers for having providing continued constructive and supportive feedback to our manuscript. Reviewers 1 and 3 have already in earlier rounds, after thorough reviews, expressed satisfaction and support for publication (“*I think this manuscript is much improved, and the authors have addressed all comments well. I recommend for publication*”, reviewer 1; “*I support its publication with some minor modifications*”, reviewer 3). Reviewer 2 has also shown increasing support, providing constructive suggestions, with some aspects on his/her suggested “test” on the dual-isotope organic carbon source apportionment now being the matter for remaining discussion.

As already explained in the last round yet elaborated further below, the reviewer’s proposed “test” to compare the output of the OC source apportionment is in practice largely unfeasible due to the lack of sensible observational constraints for all but one OC fraction, i.e. the petrogenic OC. As explained in detail below, the reviewer’s suggestion to test the OC source apportionment with “%N, %OC, C/N, or $\delta^{15}\text{N}$ is not realistic in practice for multiple reasons, but foremost because of poorly-defined and non-specific source fingerprints, as well as their extensive biogeochemical processing, disturbing these bulk-element source signals. Moreover, N clearly has inherently different sources and very contrasting biogeochemical cycling when compared to OC. We have, however, now included such a “test” based on %OC for the only OC sub-fraction for which this may work, the petrogenic OC fraction, and this test shows that the results of our dual-isotope source apportionment are in line with previous work, which was based on much smaller datasets, that focused specifically on the petrogenic OC component in geographically minor areas of the circum-Arctic region where this sub-component is relevant. In our writing below, we further elaborate why the suggested approach (the “test”) is not applicable for other OC fractions than the petrogenic component. While we have repeatedly demonstrated that the results of the petrogenic OC fraction are reasonable, we like to remind that this repeatedly emphasized (by the reviewer) OC sub-component is i) certainly not the motivation and focus of our study, ii) only relevant for a few of the shelf-sea regions, and iii) actually not included in any of the circum-Arctic assessment on terrestrial OC release from soil, permafrost and peat (mentioned in line 161, 300 and the caption of Fig. 1), which is the major contribution of this study.

The manuscript is now further revised to address this aspect and the few other minor concerns remaining from reviewer 2. We have also, on the reviewer’s request, now included Supplementary Figure 2 into the main manuscript. The comments by reviewer 2 are shown in *black italics font*, followed by our responses in blue text. Our responses refer to line numbers of the new revised manuscript version.

Reviewer #2

I include some more detail of the “test” I was proposing of their mixing analysis below, in addition to clarifying an earlier request to move the Supplementary Figure 2 into the main text of the article.

General Comment 1

Author reply - “We naturally agree with the spirit of the reviewer's comment that it is always important to clarify weaknesses and uncertainties of any approach; we are convinced the present manuscript does this, yet we seek to further improve and clarify this part. However, it is not possible to follow the suggestion to test the model output against %OC, %N or C/N because the dual-isotope source apportionment does not have any of the three parameters as output.”

Response - This is precisely my point. Because the analysis does not output these, they can be used independently to explore how the model is behaving. The key question is are the fractions of carbon from each end members re-producing sensible bulk elemental proxies? (note you could also imagine an analysis which uses $d^{15}\text{N}$ as test). So, in an ideal world, you would have a set of data which is not being used in the mixing analysis, from which the mixing outputs could be explored.

We appreciate these attempts to clarify. However, we again like to note that testing the output of the OC source apportionment using %OC, %N, C/N, or even $\delta^{15}\text{N}$ is not feasible. All these parameters are operating differently than the OC mixing reflected by $\Delta^{14}\text{C}$ and $\delta^{13}\text{C}$ (which are inherent, intrinsic properties of the actual OC) and would therefore not give useful results. First, both %N and $\delta^{15}\text{N}$ would in the best of cases allow an assessment of N sources, but N sources are inherently different than those of OC, and N is not the topic of this study. If we were to use these parameters to carry out source apportionments, we would encounter major limitations with respect to distinguishing signatures between different N sources that are further influenced by various N cycle processes that are not source-specific. More specifically, $\delta^{15}\text{N}$ is influenced by many processes that show strong isotopic fractionation (see e.g., Bedard-Haughn et al., 2003; or Denk et al., 2017 for an overview) and thus lead to highly variable $\delta^{15}\text{N}$ values of possible source endmembers. For instance, the active layer of permafrost can range between ca. 0 and 6‰ (Hugelius et al., 2012; Wild et al., 2018). Particulate N in large Arctic rivers shows high variability both between rivers and seasons, with average values between -1.4 and +6.3‰ (McClelland et al., 2016). The limitations of $\delta^{15}\text{N}$ values are especially problematic at a large spatial scale such as in our study, where reliable endmember values are very poorly constrained. Moreover, N concentrations in sediments are influenced by rates of primary production, input of land-derived N, sediment clay content that can bind inorganic N, benthic N remineralization, N_2 fixation, etc. Normalizing N by OC as OC/N ratios may initially appear more promising since marine sources have comparatively well constrained OC/N ratios around the Redfield ratio, but the OC/N ratio may be quite different for ice-associated plankton, and OC/N values of different terrestrial sources are much less constrained, highly variable and also strongly influenced by decomposition that reduces the OC/N ratio. For instance, OC/N ratios of active layer soils can vary from >30 in poorly degraded plant litter at the surface to <10 at the bottom of the active layer (Gentsch et al., 2015). Also for Ice Complex Deposits, a wide range of values from >20 to <5 has been found (Schirrmeister et al., 2011). Winterfeld et al. (2015) offers a more wide-ranging discussion of the limitations of OC/N for source apportionment.

The $\Delta^{14}\text{C}/\delta^{13}\text{C}$ approach is the only (currently-available) technique that has the strength to resolve between different (marine vs. terrestrial, and young vs. old) OC sources in coastal sediments, which is why we have repeatedly mentioned and pointed out literature, in which these isotopes were studied and critically assessed. Research has employed a wide range of terrestrial biomarkers (wax lipids, lignin phenols, GDGTs) and other isotope systems at the bulk and molecular-level ($\delta^2\text{H}$ and $\Delta^{14}\text{C}$ of wax lipids; e.g., Drenzek et al., 2007; Feng et al., 2013; Gustafsson et al., 2011; Vonk et al., 2017) with some comparing with the $\Delta^{14}\text{C}/\delta^{13}\text{C}$ -based source apportionment. However, please note that these are not “tests” – as these biomarkers are well-recognized to each trace different sub-components of the OC. The distinction between different terrOC sources is very difficult at the molecular level as most of these biomarkers originate from plant biomass, have quite variable source ratios, and represent a trace fraction of the total terrestrial OC. Further, data on biomarkers is currently not available at the circum-Arctic scale. We naturally agree and acknowledge (here and in the manuscript) that bulk OC source apportionment based on $\Delta^{14}\text{C}/\delta^{13}\text{C}$ has limitations, which are mirrored by the large uncertainties our estimates have, yet it is the by far best approach that currently exists as the $\Delta^{14}\text{C}$ and $\delta^{13}\text{C}$ isotope ratios are intensive properties of the OC itself. We believe that limitations now are sufficiently explained throughout the manuscript (e.g. line 88-101, 152-159, 312-339).

Author reply - “Given the lack of any further suggestion from the reviewer how this should be done, it seems likely that this suggestion builds on a misunderstanding of our method as the approach suggested by the reviewer is simply not feasible.”

Response - My apologies for not expanding. The underlying maths is straightforward and similar to those underpinning the approach applied in the paper, so I thought it would clear what I was getting at. Thinking about the organic carbon concentration example, which I imagine you should have the data in your compilation (although it is not in the data files), then we can write that f_a – which is the fraction of carbon from end member a (an output of the dual isotope mixing analysis) – is equal to:

$$f_a = [\text{OC}]_a / [\text{OC}]_{\text{tot}}$$

where $[OC]_{tot}$ is the total organic carbon concentration in the sample being unmixed, while $[OC]_a$ is the organic carbon concentration in the sample mass brought by end member a (note I have skipped a step in the algebra involving the conversion of mass of carbon to concentration of carbon).

So, if we output f_a , we can quickly calculate a predicted $[OC]_a$, just knowing the organic carbon concentration in the sample.

For this study, a logical component to look at as a “test” would be the petrogenic component, as that $[OC]\%$ will reflect a pre-mixture of organic matter and mineral matrix, and rock organic carbon concentrations are measured a lot for other reasons. One can calculate the expected $[OC]_{petro}$, and see if it makes sense. If its high, perhaps the aged OM end member is more important than the mixing model suggests based on the input end member. Or, perhaps the aging during transport or residence at the seafloor has been underestimated.

As an example, one could look at the BFS samples listed in Supplementary Table 5. Just picking a sample here to show what I mean, PG2318-1 (where the $[OC]_{tot}$ is reported in a master thesis online, Riedel, 2017, as $[OC]_{tot} = 1.3\%$) has a $f_{petro} = 0.52$. This suggests a $[OC]_{petro} = 0.65\%$. For PG2313-1, $[OC]_{tot} = 2.1\%$ and f_{petro} is reported here = 0.60. This suggests a $[OC]_{petro} = 1.26\%$. These do seem high, the second one especially, when they are compared to the rivers in the region (refs 35 and 36 in the manuscript). One could plot these predicted $[OC]_{petro}$ for different sites, and see if the numbers makes sense.

Overall, I hope this quick analysis is easy to follow for the authors and clarifies what I meant in my last 2 reviews. It suggests a petrogenic component may be being overestimated and an old soil organic matter component underestimated. This could mean the Beaufort Sea circles and %surface in the figures have more uncertainty?

I haven't done this wider in the dataset, but the petrogenic end member check could also be used for the Barents Sea.

The mathematical equation described by the reviewer is certainly straightforward. However, there are major challenges; i) the parameterization of the “end member” of each fraction; and ii) the assumption that no processing occurs between the land-based source and shelf sediment receptor. Huge uncertainties (often orders of magnitude) of these two major challenges undermine this suggested “test” approach for almost all components; such that it is not feasible to “test” whether the results of the dual-isotope source apportionment “make sense” or not.

Comparing %OC of any fraction in marine sediments with other matrices (such as fluvially-suspended particulate OC, OC in coastal permafrost deposits, or OC in soils and peat) remains extremely challenging, first and foremost because these other matrices hold large uncertainties. For fluvial organic matter, OC concentrations and different source fractions are poorly constrained, vary considerably throughout seasons (McClelland et al., 2016), and existing findings often rest on small datasets. For all deposit-like compartments (soil, permafrost and peat), OC concentrations vary by two orders of magnitude (Schirmeister et al., 2011; Hugelius et al., 2013). On top of these large uncertainties, there are dilution and sediment matrix effects that influence OC concentrations in marine environments, which further complicate comparing bulk %OC concentrations between sediments and other matrices and sources.

The reviewer suggests “as an example” to look at the petrogenic OC fraction. We agree to follow the reviewer’s suggestion for this fraction because petrogenic OC is the only OC fraction for which it may be reasonable to do this calculation. The reason is that petrogenic OC is the only fraction that presents relatively narrow boundaries of %OC from petrogenic sources ($\%OC_{petro}$), for which thus a comparison between $\%OC_{petro}$ in sediments and other matrices may be reasonable.

Calculating the $\%OC_{petro}$ for Beaufort Sea sediments (east of 140°W) reveals a concentration of $0.66 \pm 0.18\%$, with a range from 0.05 to 1.26% (the maximum value calculated by the reviewer). As discussed in detail in our first author response, these values are in a similar range as a large variety of $\%OC_{petro}$ concentrations reported in the literature. For instance, previous $\%OC_{petro}$ estimates of fluvially-

suspended material in the Beaufort Sea catchment are 0.12-0.63% (Hilton et al., 2015) and 0.26% (Vonk et al., 2015). We also like to mention that Mackenzie river export is not the only source of OC_{petro}, as additional OC_{petro} may be imported from other smaller basins or remobilized from coastal permafrost that also contains petrogenic material (Bröder et al., 2021).

While a comparison of relative OC_{petro} fractions with prior research was added to the manuscript already in the first revision, we have now amended this part to also include the direct “test”/comparison based on %OC_{petro} concentrations:

“Further, the concentrations of petrogenic OC in Beaufort Sea sediments (0.66±0.18%; mean±s.d.) are also similar to the range of petrogenic OC concentrations in fluvial suspended material from within the Mackenzie basin (0.12-0.63%)(Hilton et al., 2015; Vonk et al., 2015), which thus is broadly consistent with the results of the dual-isotope source apportionment and indicates that petrogenic OC in Beaufort Sea sediments is largely attributed to rock weathering and river export from the Mackenzie river and its tributaries (Hilton et al., 2015).” (lines 154-159)

Response to “General Comment 3”

The small clarifications in the main text are welcome.

However, please put Figure S2 in the main text... For a paper like this in Nat comms, most readers won't understand fully what data has been used to produce the figures or the “index” here. For me, that is a problem, yet the fix is really straightforward – Supplementary Figure 2 becomes a main text figure. These are the raw measurements which we have confidence in. The figure also shows the compilations of the end members very nicely. Finally, it shows the decision making by the authors that have selected different end members for each basin. It really is an important figure to have in the main paper, not hidden away.

We can agree with the argument that this figure would make it easier for the reader to follow the selection of the different end members and the underlying $\Delta^{14}\text{C}/\delta^{13}\text{C}$ data points in each shelf sea. After careful re-consideration of this comment we decided to follow the reviewer's recommendation and thus moved Supplementary Fig. 2 to the main manuscript (now Fig. 2).

Specific review comments 1b & c (on potential OM aging):

A quick look on a reference search finds reported Barents Sea sedimentation rates from 0.004 to 0.21 cm/yr, average 0.06 cm/yr (compiled in the publication Faust et al., 2020, Philosophical Transactions of the Royal Society A). Looking at the figures in that publication, a number of sites away from coastal settings have sedimentation rates <0.01 cm/yr. So in that context, the upper 2 cm could have shifts in $\Delta^{14}\text{C}$ of approximately ~30 permille due to aging. If is the extreme of that range (0.004 cm/yr), it would be a shift of about ~70 permille. Is this worth noting? 0.21+/-0.22 is quite a range, and clearly the lower sedimentation rate sites could be impacted by in-situ aging?

As already addressed previously we can rule out that in-situ aging has any relevant effect on the OC source apportionment. We like to stress that the extreme examples by the reviewer of $\Delta^{14}\text{C}$ depletion of 30‰ at a sedimentation rate of 0.01 cm/yr, and 70‰ at 0.004 cm/yr, are incorrect as these values refer to the depletion at 2 cm depth and not $\Delta^{14}\text{C}$ depletion integrated over the 0-2 cm depth interval. Following the reviewer's example choice of (unusually low) sedimentation rates, the correct depletions for a sample from the 0-2 cm interval would be 9 and 39‰, respectively.

Nonetheless, to further demonstrate that neither of these depletions have a large effect we have now re-run the source apportionment calculations for the Beaufort and Barents Sea samples. The first scenario tests this for the average $\Delta^{14}\text{C}$ depletion of the 0-2 cm depth interval at the more tangible circum-Arctic average sedimentation rate of 0.21 cm/yr. The second scenario then applies the 39‰ depletion under the exceptionally low sedimentation rate of 0.004 cm/yr (note that the Barents Sea average is 0.12

cm/yr!). The different sedimentation rates and resulting residence times translate into a $\Delta^{14}\text{C}$ shift of the marine OC end member from -50‰ to -59‰ and -89‰, respectively. Based on the average sedimentation rate of circum-Arctic shelf sediments (0.21 cm/yr), the resulting OC fractions for samples from both the Beaufort and Barents seas are shifting with <1% (of the total OC) and are indistinguishable from the case where the regular marine OC end member (-50‰) is applied. Under extremely low sedimentation rates (0.004 cm/yr), the questioned petrogenic OC fraction shrinks only by $0.9 \pm 0.6\%$ of the OC in Beaufort Sea sediments and by $1.4 \pm 0.9\%$ in Barents Sea sediments, while marine OC fractions increase by about 1%. In either case, the difference is much smaller than the large estimated uncertainty around the mean estimated petrogenic OC fraction ($15 \pm 10\%$ of the total OC in the Barents Sea and $51 \pm 13\%$ for the Beaufort Sea). This demonstrates that in-situ aging, even under extremely low sedimentation rates, can clearly not explain the large proportion of pre-aged or petrogenic OC. Moreover, most data used by this study is based on samples and ^{14}C -OC measurements from the 0-1 cm interval and not of 0-2 cm. Another important point is that the key shelf seas that are the most important receptors of remobilized permafrost and peat OC, all have notably higher average sedimentation rates than the Barents Sea (Supplementary Fig. 2).

Please note that a line about the unlikeliness of in-situ aging to affect the source apportionment was added to the manuscript during the last revision stage, which we have now further revised to also add the results of this new sensitivity analysis:

“We further tested the effect of any post-depositional OC aging to the source apportionment. Based on the average ^{210}Pb sedimentation rates in circum-Arctic shelf sediments (0.21 ± 0.22 cm/yr average for this study; $n=164$), and the corresponding ^{14}C depletion of marine OC over the 0-2 cm depth interval, we find resulting OC fractions to shift <1% (of the total OC) compared to source fractions for which the end member remained unchanged. Even under extremely low sedimentation rates (0.004 cm/yr), the (oldest) petrogenic OC fraction in Beaufort and Barents seas sediments is only around 1% smaller, which is much lower than the uncertainty estimates of this fraction. Hence, any post-depositional OC aging can be ruled out and will not significantly affect the source apportionment.” (line 331-339)

References

- Bedard-Haughn, A., van Groenigen, J. W. and van Kessel, C.: Tracing ^{15}N through landscapes: potential uses and precautions, *J Hydrol (Amst)*, 272(1–4), 175–190, doi:10.1016/S0022-1694(02)00263-9, 2003.
- Bröder, L., Keskitalo, K., Zolkos, S., Shakil, S., Tank, S. E., Kokelj, S. v, Tesi, T., van Dongen, B. E., Haghypour, N., Eglinton, T. I. and Vonk, J. E.: Preferential export of permafrost-derived organic matter as retrogressive thaw slumping intensifies, *Environmental Research Letters*, 16(5), 054059, doi:10.1088/1748-9326/abee4b, 2021.
- Denk, T. R. A., Mohn, J., Decock, C., Lewicka-Szczebak, D., Harris, E., Butterbach-Bahl, K., Kiese, R. and Wolf, B.: The nitrogen cycle: A review of isotope effects and isotope modeling approaches, *Soil Biol Biochem*, 105, 121–137, doi:10.1016/J.SOILBIO.2016.11.015, 2017.
- Drenzek, N. J., Montluçon, D. B., Yunker, M. B., Macdonald, R. W. and Eglinton, T. I.: Constraints on the origin of sedimentary organic carbon in the Beaufort Sea from coupled molecular ^{13}C and ^{14}C measurements, *Mar Chem*, 103(1–2), 146–162, doi:10.1016/j.marchem.2006.06.017, 2007.
- Feng, X., Vonk, J. E., van Dongen, B. E., Gustafsson, Ö., Semiletov, I. P., Dudarev, O. v, Wang, Z., Montluçon, D. B., Wacker, L. and Eglinton, T. I.: Differential mobilization of terrestrial carbon pools in Eurasian Arctic river basins., *Proc Natl Acad Sci U S A*, 110(35), 14168–73, doi:10.1073/pnas.1307031110, 2013.
- Gentsch, N., Mikutta, R., Alves, R. J. E., Barta, J., Čapek, P., Gittel, A., Hugelius, G., Kuhry, P., Lashchinskiy, N., Palmtag, J., Richter, A., Šantručková, H., Schneckner, J., Shibistova, O., Urich, T., Wild, B. and Guggenberger, G.: Storage and transformation of organic matter fractions in cryoturbated permafrost soils across the Siberian Arctic, *Biogeosciences*, 12(14), 4525–4542, doi:10.5194/bg-12-4525-2015, 2015.
- Gustafsson, Ö., van Dongen, B. E., Vonk, J. E., Dudarev, O. v and Semiletov, I. P.: Widespread release of old carbon across the Siberian Arctic echoed by its large rivers, *Biogeosciences*, 8(6), 1737–1743, doi:10.5194/bg-8-1737-2011, 2011.
- Hilton, R. G., Galy, V., Gaillardet, J., Dellinger, M., Bryant, C., O'Regan, M., Gröcke, D. R., Coxall, H., Bouchez, J. and Calmels, D.: Erosion of organic carbon in the Arctic as a geological carbon dioxide sink, *Nature*, 524(7563), 84–87, doi:10.1038/nature14653, 2015.
- Hugelius, G., Routh, J., Kuhry, P. and Crill, P.: Mapping the degree of decomposition and thaw remobilization potential of soil organic matter in discontinuous permafrost terrain, *J Geophys Res Biogeosci*, 117(G2) [online] Available from: <http://doi.wiley.com/10.1029/2011JG001873>, 2012.
- Hugelius, G., Tarnocai, C., Broll, G., Canadell, J. G., Kuhry, P. and Swanson, D. K.: The Northern Circumpolar Soil Carbon Database: spatially distributed datasets of soil coverage and soil carbon storage in the northern permafrost regions, *Earth Syst Sci Data*, 5(1), 3–13 [online] Available from: <http://www.earth-syst-sci-data.net/5/3/2013/>, 2013.
- McClelland, J. W., Holmes, R. M., Peterson, B. J., Raymond, P. A., Striegl, R. G., Zhulidov, A. v., Zimov, S. A., Zimov, N., Tank, S. E., Spencer, R. G. M., Staples, R., Gurtovaya, T. Y. and Griffin, C. G.: Particulate organic carbon and nitrogen export from major Arctic rivers, *Global Biogeochem Cycles*, 30(5), 629–643, doi:10.1002/2015GB005351, 2016.
- Schirrmeister, L., Kunitsky, V., Grosse, G., Wetterich, S., Meyer, H., Schwamborn, G., Babi, O., Derevyagin, A. and Siegert, C.: Sedimentary characteristics and origin of the Late Pleistocene Ice Complex on north-east Siberian Arctic coastal lowlands and islands - A review, *Quaternary International*, 241(1–2), 3–25, doi:10.1016/j.quaint.2010.04.004, 2011.

Vonk, J. E., Tesi, T., Bröder, L., Holmstrand, H., Hugelius, G., Andersson, A., Dudarev, O., Semiletov, I. and Gustafsson, Ö.: Distinguishing between old and modern permafrost sources in the northeast Siberian land–shelf system with compound-specific δ 2H analysis, *Cryosphere*, 11(4), 1879–1895, doi:10.5194/tc-11-1879-2017, 2017.

Vonk, J. E., Giosan, L., Blusztajn, J., Montlucon, D., Graf Pannatier, E., McIntyre, C., Wacker, L., Macdonald, R. W., Yunker, M. B. and Eglinton, T. I.: Spatial variations in geochemical characteristics of the modern Mackenzie Delta sedimentary system, *Geochim Cosmochim Acta*, 171, 100–120, doi:10.1016/j.gca.2015.08.005, 2015.

Wild, B., Alves, R. J. E., Bárta, J., Čapek, P., Gentsch, N., Guggenberger, G., Hugelius, G., Knoltsch, A., Kuhry, P., Lashchinskiy, N., Mikutta, R., Palmtag, J., Prommer, J., Schneckner, J., Shibistova, O., Takriti, M., Urich, T. and Richter, A.: Amino acid production exceeds plant nitrogen demand in Siberian tundra, *Environmental Research Letters*, 13(3), 034002, doi:10.1088/1748-9326/aaa4fa, 2018.

Winterfeld, M., Laepple, T. and Mollenhauer, G.: Characterization of particulate organic matter in the Lena River delta and adjacent nearshore zone, NE Siberia - Part I: Radiocarbon inventories, *Biogeosciences*, 12(12), 3769–3788, doi:10.5194/bg-12-3769-2015, 2015.